# XPF activates break-induced telomere synthesis

Chia-Yu Guh[1,8], Hong-Jhih Shen[1,8], Liv WeiChien Chen[1,8], Pei-Chen Chiu[1,8], I-Hsin Liao[1], Chen-Chia Lo[1], Yunfei Chen[1], Yu-Hung Hsieh[1], Ting-Chia Chang[1], Chien-Ping Yen[1], Yi-Yun Chen [2], Tom Wei-Wu Chen [3], Liuh-Yow Chen [4], Ching-Shyi Wu[5], Jean-Marc Egly[6,7] & Hsueh-Ping Catherine Chu [1] ✉

Alternative Lengthening of Telomeres (ALT) utilizes a recombination mechanism and break-induced DNA synthesis to maintain telomere length without telomerase, but it is unclear how cells initiate ALT. TERRA, telomeric repeat-containing RNA, forms RNA:DNA hybrids (R-loops) at ALT telomeres. We show that depleting TERRA using an RNA-targeting Cas9 system reduces ALT-associated PML bodies, telomere clustering, and telomere lengthening. TERRA interactome reveals that TERRA interacts with an extensive subset of DNA repair proteins in ALT cells. One of TERRA interacting proteins, the endonuclease XPF, is highly enriched at ALT telomeres and recruited by telomeric R-loops to induce DNA damage response (DDR) independent of CSB and SLX4, and thus triggers break-induced telomere synthesis and lengthening. The attraction of BRCA1 and RAD51 at telomeres requires XPF in FANCM-deficient cells that accumulate telomeric R-loops. Our results suggest that telomeric R-loops activate DDR via XPF to promote homologous recombination and telomere replication to drive ALT.

Continuous lengthening of the telomeres is one of the hallmarks of cancers. Most cancers require telomerase activity to maintain the telomere length to achieve immortality. Nevertheless, a subset of cancers depends on a mechanism, which is called Alternative Lengthening of Telomeres (ALT), to extend telomere length independent of telomerase[1–4] by exploiting homologous recombination[5,6] and break-induced DNA synthesis[7,8]. ALT pathway has been observed in various types of human tumors such as sarcomas and gliomas[9,10]. Sarcomas that utilize ALT carry a higher risk of patient death compared to telomerase-positive tumors[11]. ALT cancer cells share some characteristics, including clustering of multiple telomeres into ALT-associated PML bodies (APBs)[12–16], elevated levels of the telomeric-repeat containing long noncoding RNA (TERRA)[17–21] and the appearance of a large

amount of extrachromosomal telomeric repeats, such as C-circles and G-circles[22,23]. Multiple DNA metabolism pathways are involved in the maintenance of telomere length in ALT cells. Break-induced replication, a conservative DNA synthesis-based repair pathway[5,7,24], is active at ALT telomeres[8,25]. The presence of replication stress and DNA damage response proteins in APBs at ALT telomeres trigger DNA repair[26] and homologous recombination[27]. However, how DNA damage responses have been provoked at ALT telomeres has not been fully understood.

Telomeric ends synthesize a heterogeneous population of long noncoding RNAs called TERRA[17], which is transcribed from subtelomeric regions toward each end of chromosomes. TERRA is transcribed by RNA polymerase II[18] and can form RNA:DNA hybrids (R-

[1]Institute of Molecular and Cellular Biology, National Taiwan University, No. 1 Sec. 4 Roosevelt Road, Taipei, Taiwan. [2]Institute of Biological Chemistry, Academia Sinica, Taipei, Taiwan. [3]Department of Oncology, National Taiwan University Hospital and Graduate Institute of Oncology, National Taiwan University College of Medicine, Taipei, Taiwan. [4]Institute of Molecular Biology, Academia Sinica, Taipei, Taiwan. [5]Department of Pharmacology, National Taiwan University, Taipei, Taiwan. [6]Department of Functional Genomics and Cancer, IGBMC, CNRS/INSERM/University of Strasbourg, Strasbourg, France. [7]College of Medicine, National Taiwan University, Taipei, Taiwan. [8]These authors contributed equally: Chia-Yu Guh, Hong-Jhih Shen, Liv WeiChien Chen, Pei-Chen Chiu. ✉e-mail: cchu2017@ntu.edu.tw

loops) at telomeres. Lines of evidence have shown that the formation of R-loops at telomeres could be one of the mechanisms to trigger DNA repair to lengthen telomeres[28–31]. Suppressing R-loop formation by overexpressing endoribonuclease RNase H1, which catalyzes the cleavage of RNA in an RNA:DNA hybrid, reduces telomere recombination[30], suggesting that R-loops are required for telomere maintenance in ALT cancer cells. The downside of these approaches is that manipulation of RNase H1 alters all RNA:DNA hybrids across the genome. A study from the Azzalin laboratory assessed the impact of TERRA transcription by targeting the 29 bp repeats of subtelomeric sequences with transcription repressors to suppress TERRA transcription in a subset of telomeres and showed that TERRA transcripts contribute to DNA damage at telomeres and break-induced telomere synthesis[32]. However, how TERRA transcription or telomeric R-loops trigger DNA damage at telomeres remains elusive.

In this study, we show that TERRA activates DNA damage response (DDR) by recruiting XPF, a nucleotide excision repair factor with endonuclease activity[33]. Mammalian XPF and ERCC1 form a stable heterodimer, which is responsible for the incision of a DNA lesion during nucleotide excision repair (NER), and the repair of DNA inter-strand crosslinks (ICLs) and double-strand breaks (DSBs)[33]. XPF contains a conserved nuclease domain for its catalytic activity, and ERCC1 is responsible for DNA binding[34]. Deficiency of XPF-ERCC1 in humans causes xeroderma pigmentosum with hypersensitivity to UV damage[33], Cockayne syndrome[35], Fanconi anemia[36], XFE progeroid syndrome[37], or similar segmental progeroid syndromes with severe features of accelerated aging[38]. A fraction of XPF–ERCC1 is associated with telomeres, where it is required for the recombination of deprotected telomeres after TRF2 overexpression[39,40]. Two structure-specific flap endonucleases, XPF and XPG are responsible for the repair of bulky lesions in the NER pathway. Evidence has shown that purified XPF-ERCC1 and XPG are capable of cleaving an R-loop structure formed at switch regions of the immunoglobulin locus in vitro[41].

To study the function of TERRA in ALT cells, we investigate the effect of TERRA depletion by using an RNA-targeting Cas9 system (RCas9)[42] to downregulate TERRA levels and identify TERRA-interacting proteins by performing iDRiP (identification of direct RNA-interacting proteins)[43]. We reveal that TERRA is crucial for telomere lengthening in ALT cells and TERRA interacts with proteins involved in DNA repair pathways including double-strand breaks, homologous recombination (HR), and nucleotide excision repair. Among TERRA interacting proteins, we find that XPF is required for DDR to promote HR and break-induced telomere synthesis in ALT cells. Ablation of XPF function shortens telomere length and suppresses cell growth in ALT cells. Our data infer a pivotal role of XPF and TERRA in the onset of ALT.

## Results

### TERRA depletion reduces alternative lengthening of telomeres

To specifically knockdown TERRA RNA in ALT cells without manip-ulating telomeric DNA, we utilized an RCas9, which successfully eliminated tri-nucleotide repetitive RNAs in the nucleus[42]. The RCas9 system carrying dCas9 fused to the PIN RNA endonuclease domain of SMG6[42] enables the cleavage of RNA after single guide RNA (sgRNA) targeting (Fig. 1a). We designed four sgRNAs for the RCas9 system: TERRA-1, TERRA-2, sense, and λ2. TERRA-1 and TERRA-2 pro-duce antisense TERRA sgRNA to guide PIN-RCas9 to TERRA RNA for degradation. Sense contains G-rich telomeric repeat sequences and λ2 generates control sgRNA that should not target TERRA. Transient transfection of RCas9-sgTERRA-1 and RCas9-sgTERRA-2 displayed a reduction of TERRA levels compared to sense and λ2 controls (Supplementary Fig. 1a, b), and RCas9-sgTERRA-1 showed higher knockdown efficiency than RCas9-sgTERRA-2. We screened and selected stable cell lines carrying RCas9-sgTERRA-1 with lower TERRA expression by RNA slot blotting (Supplementary Fig. 1c). Northern blot

analysis and RNA fluorescence in situ hybridization (FISH) indicated the decreased TERRA expression in cell lines carrying RCas9-sgTERRA-1 (sgTERRA_1C6 and sgTERRA_1C21) (Fig. 1b–d), in comparison to RCas9-sgSense and RCas9-sgλ2 controls.

TERRA knockdown resulted in the reduction of ALT features including large TRF2 foci (top 5% TRF2 intensity) and APB formation, which was determined by colocalization of PML and large TRF2 foci (Fig. 1d–f). To monitor telomeric R-loops, we performed DNA:RNA immunoprecipitation (DRIP) using S9.6 antibody against RNA:DNA hybrids. The quantitative PCR demonstrated a reduction of telomeric R-loops in TERRA knockdown cells (Fig. 1g). Long-term down-regulation of TERRA in RCas9-sgTERRA cells allowed us to assess the consequences of telomere lengthening in ALT cells. Interestingly, the Telomere Restriction Fragment (TRF) analysis revealed that TERRA knockdown cells at PD6 (population doubling number 6) contained a shortened telomere length (Fig. 1h). Our results indicate that TERRA promotes APB formation and telomere lengthening in ALT cells.

### TERRA contributes to telomere clustering

PAR-TERRA, a telomeric repeat-containing RNA derived from the pseudoautosomal regions of sex chromosomes, was thought to direct the clustering of the sex chromosome ends[44], thus creating a hub to constrain the DNA loci in 3D space and promoting transient homo-logous pairing during embryonic stem cell differentiation. We then asked whether TERRA could mediate telomere clustering, an early event to promote APB formation to drive ALT[45]. To analyze telomere clustering events, the intensity and the size of telomere foci were quantified after telomere DNA FISH using peptide nucleic acid probes (Fig. 2a–e, Supplementary Fig. 2a). We quantified the telomere inten-sity of each spot (Fig. 2a–c), and selected foci with the highest telo-meric intensity (top 5%) as telomere clustering events, in which several telomeres interact with each other (Fig. 2a, d, e). The top 5% of telo-mere intensity was around 6-10 fold higher (1976 a.u. in control sgλ2 and 1519 a.u. in sgTER1C6 cells) than the median of telomere intensity (244 a.u. in control sgλ2 and 151 a.u. in sgTER1C6 cells). The largest telomere foci, which are over 1.2 μm$^2$ in size, also account for about 5% of the population (Supplementary Fig. 2a) in control cells. The quan-tification data indicated that TERRA depletion reduced the intensity (Fig. 2b–d) and the size (Fig. 2e, Supplementary Fig. 2a) of telomere clustering events. This result is consistent with the observation in TERRA knockdown cells containing reduced large TRF2 foci, which usually mark clustered telomeres (Fig. 1e).

Given that the Bloom helicase (BLM) and Replication Protein A (RPA) were found to interact with TERRA[43], we asked if TERRA alters BLM and RPA localization. Notably, TERRA depletion resulted in the reduction of the colocalization of BLM and large TRF2 foci (Fig. 2f, g). Similarly, the colocalization events of RPA70 and large TRF2 foci were decreased in TERRA knockdown cells (sgTER1C6, sgTER1C21) (Fig. 2h, Supplementary Fig. 2b). When using RAP1 (repressor/acti-vator protein 1) as a mark for telomere, the data were consistent and showed the reduction of colocalization of BLM and large RAP1 in TERRA knockdown cells (Supplementary Fig. 2c). These results imply that TERRA promotes the recruitment of BLM and RPA to ALT telo-meres. We also observed that BLM depletion decreased the top 5% telomere intensity in parental U2OS cells (Supplementary Fig. 2d, e), supporting the idea of BLM promoting telomere clustering and inducing ALT activity[46,47]. When BLM was depleted in RCas9-sgRNA cell lines, the top 5% telomere intensity was significantly reduced (Fig. 2i, Supplementary Fig. 2f). The BLM depletion in TERRA knockdown cells exhibited the lowest top 5% telomere intensity (Fig. 2i) among other control cell lines (RCas9-sgSense and RCas9-sgλ2), indicating that both TERRA and BLM facilitate telomere clus-tering. To examine whether R-loops contribute to telomere cluster-ing, RNase H1-mCherry was exogenously expressed in U2OS cells (Supplementary Fig. 2g). Cells exogenously expressing wildtype

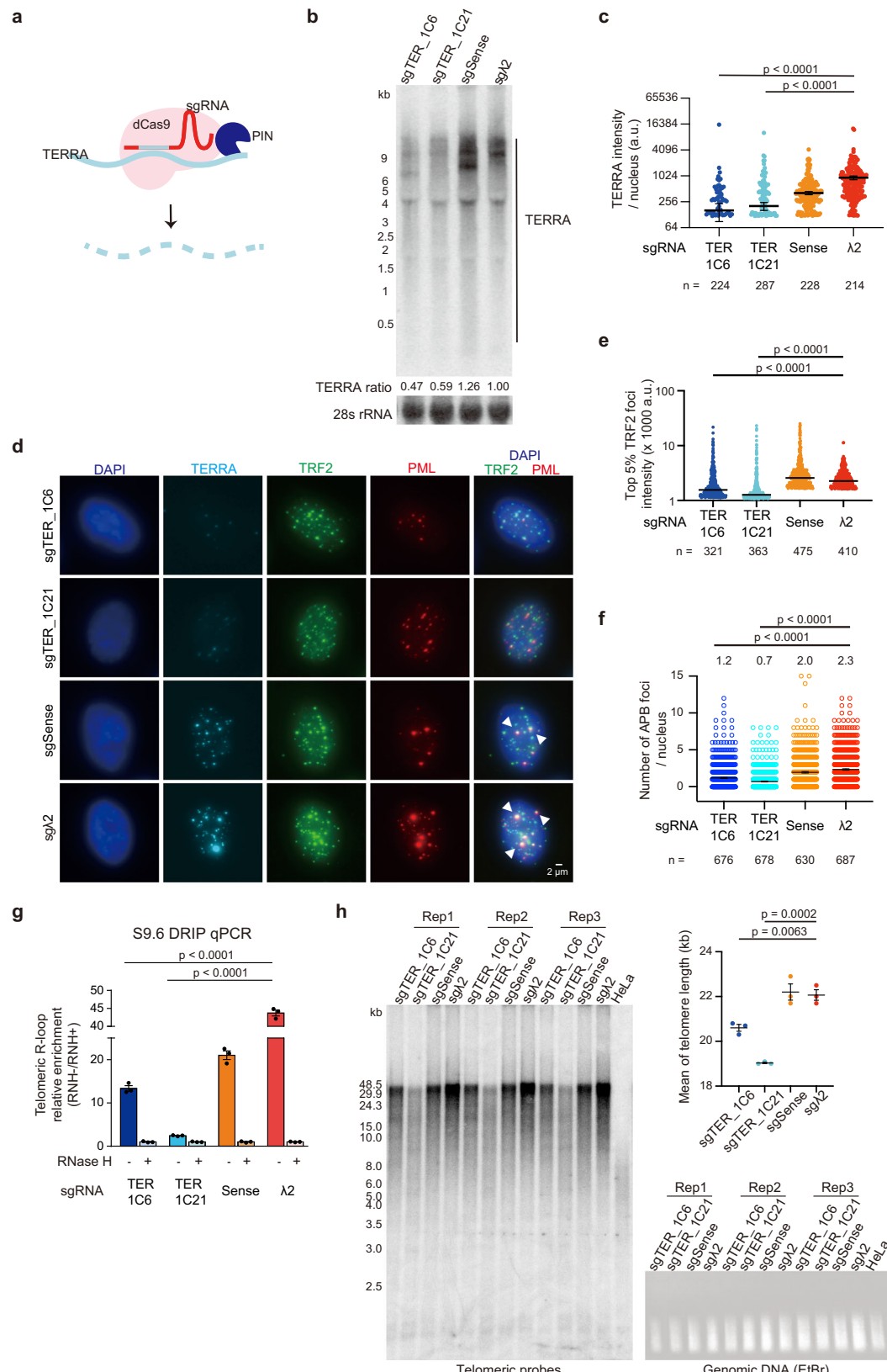

RNase H1-mCherry displayed a reduction of telomeric R-loops and telomere clustering (Supplementary Fig. 2h, i), in comparison to cells expressing catalytic-dead mutant RNase H1 or vector alone. Moreover, silencing endogenous RNase H1 increased telomere clustering, APBs, as well as the recruitments of BLM and RPA70 to ALT telomeres (Fig. 2j–n). Collectively, these data suggest that TERRA and its associated R-loops (TERRA R-loops) facilitate the recruitment of BLM and RPA70 to ALT telomeres and promote telomere clustering.

## TERRA interacts with DNA repair proteins in ALT cancer cells

We next sought to identify TERRA-interacting proteins in U2OS cells by performing iDRiP, a method to capture specific RNA interacting

**Fig. 1 | TERRA depletion by the RCas9 system reduces alternative lengthening of telomeres. a** RCas9-sgRNA system to deplete TERRA RNA. The RCas9 system contains dCas9 fused to PIN RNA endonuclease. **b** Northern blot analysis showing TERRA levels in TERRA knockdown U2OS cells (sgTER_1C6, sgTER_1C21) or control cells (sgSense, sgλ2). **c** Quantification of TERRA intensity in (**d**). *P*-values, two-sided Mann-Whitney U-test. Bars, mean ± SEM. n, cell number. **d** Immuno-RNA FISH to detect TERRA RNA, TRF2, and PML in U2OS cells expressing RCas9-sgRNAs. Arrowheads indicate APBs containing big TRF2 foci colocalized with PML. **e** Quantification of top 5% TRF2 foci intensity in (**d**) using 3D images by Imaris software. Bars, median. *n* = number of TRF2 foci. Data of three independent experiments. *P*-values, two-sided Mann-Whitney U test. **f** Quantification of APB foci in TERRA knockdown cells (sgTER_1C6, sgTER_1C21) or control cells (sgSense,

sgλ2). APB foci were determined by the extensive TRF2 staining (> top 5% of TRF2 foci intensity in the control cells) with PML staining. Bars, mean ± SEM. *n*, cell number. Data of three independent experiments. *P*-values, two-sided Mann-Whitney U-test. Mean values of each group shown on the top of figures. **g** Quantification of telomeric R-loops by qPCR after DNA-RNA immunoprecipitation using S9.6 antibody. *P*-values, two-tailed student's *t*-test. bars, mean ± SEM. **h** Telomere length measured by TRF analysis in cells expressing RCas9-sgRNAs at PD6 after selection. The same amount of DNA as TRF assay was loaded in the EtBr gel. The mean of telomere length was quantified (top-right). *P*-values by two-tailed Student's *t*-test. Bars, mean ± SEM. Three independent experiments were averaged. **c**, **e**, **g** Representative of three independent experiments. Other replicates show similar trends and are provided in the Source Data file.

proteins by UV light crosslinking using antisense probe capture[43,48]. We designed probes targeting UUAGGG repeats of TERRA and several control probes, including sense (TERRA-reverse complement), luciferase (Luc, a negative control), and the mammalian U1 spliceosomal RNA (a positive control). After TERRA-iDRiP capture, RNA slot blotting and qRT-PCR showed above 30% recovery (Fig. 3a, b) and high specificity, with up to a 9000-fold enrichment compared to Luc control. TERRA-capture iDRiP yielded a high signal-to-noise ratio (normalized with U6 RNA), with a 3450-fold enrichment of TERRA over Luc control (Supplementary Fig. 3a). To obtain the profile of TERRA interacting proteins, quantitative mass spectrometry (MS) was conducted for four biological replicates. We reported 102 TERRA interacting proteins with p values smaller than 0.02 and at least > 1.5-fold enrichment over Luc control (Fig. 3c, d). The major U1 snRNPs including snRNP70 and U1 snRNP A were on the top 5 enriched hits in both U1-iDRiP MS in human and mouse cells (Supplementary Fig. 3b, c), indicating the qualified iDRiP-MS experiments. Comparing TERRA-iDRiP-MS data from mouse ES cells[43] and ALT cells, several proteins including POT1, CTC1, PML, BLM, FANCA, RPA1 (RPA70) and RPA2 (RPA32) were enriched in common (Supplementary Fig. 3d). Importantly, TERRA interacts with proteins that are involved in several DNA repair pathways including nucleotide excision repair (NER), base-excision repair, and DNA double-strand break (DSB) repair in ALT cancer cells (Fig. 3d). Regulators of homologous recombination repair including BRCA1, BRIP1, RAD50, WRN, WRNIP1, ATR, and ATRIP were also enriched. Interestingly, the endonuclease XPF, a key component regulating the NER pathway, was on the top 20 list of the highest enriched scores in TERRA iDRIP-MS (Fig. 3c). Fanconi anemia complementation group M (FANCM), which is capable of unwinding TERRA R-loops[49], was enriched in TERRA-iDRiP-MS (Fig. 3d). Gene ontology (GO) algorithm revealed that TERRA-iDRiP enrichment for factors involved in telomere maintenance, DNA replication, p53 signaling, interstrand cross-link repair, and double-strand break repair via homologous recombination (Fig. 3e). The proteomic data imply an important role of TERRA in DNA damage response at ALT telomeres.

## TERRA R-loops trigger XPF localization to telomeres

Several factors involved in the NER pathway such as XPF and general transcription factor II H (GTF2H1) were enriched in TERRA-iDRiP-MS (Fig. 3c, d) in ALT cancer cells but not in mouse embryonic stem cells, implying that NER factors have specific functions at ALT telomeres. Ultraviolet light cross-linking-RNA immunoprecipitation (UV-RIP) further confirmed the interaction of XPF and TERRA in ALT cells (Supplementary Fig. 3e). Immunostaining results showed that the colocalization of XPF and TRF2 was significantly higher in ALT positive cells (U2OS and WI38-VA cells) compared to non-ALT cells (HeLa cells) (Fig. 4a, b). About 50% of TRF2 foci co-stained with XPF in U2OS and WI38-VA, whereas only 24% of TRF2 foci co-stained with XPF in HeLa cells. Likewise, other non-ALT cell lines such as HT-1080 and SK-LMS-1 showed lower colocalization of TRF2 and XPF, in comparison to ALT positive cells (Fig. 4b). Given that ALT telomeres are less compacted[20]

and enriched with TERRA transcription and R-loops[30], we tested whether these features trigger XPF recruitment to telomeres. Notably, the ablation of R-loop formation by exogenously expressing wildtype RNase H1 (Fig. 4a, b) or depleting TERRA expression decreased the colocalization of XPF and TRF2 (Fig. 4c, Supplementary Fig. 4a), suggesting that TERRA R-loops promote the recruitment of XPF to telomeres.

## XPF contributes to DNA damage response at ALT telomeres

XPF is an endonuclease that enables cutting off R-loop-duplex junctions in both strands and generates DNA double-strand breaks (DSBs)[41,50] in vitro. We asked if XPF acts on TERRA R-loops at ALT telomeres to produce DSBs. Silencing XPF using two siRNAs exhibited a significant reduction of γH2AX foci at telomeres (Fig. 4d–f, Supplementary Fig. 4b) in U2OS cells. RPA at telomeres was also reduced in XPF depleted cells (Fig. 4g, h). Moreover, the depletion of XPG, another NER endonuclease, or XPG/XPF double-knockdown resulted in the decrease of γH2AX and RPA foci at telomeres (Supplementary Fig. 4c–f, 5a, b), indicating that XPG in addition to XPF promotes DNA damage response (DDR) at ALT telomeres.

Evidence showed that FANCM can unwind telomeric R-loops[49], and loss of FANCM accumulates R-loops and induces massive DDR at ALT telomeres[51,52]. We speculated that XPF could mediate DSBs induced by accumulation of R-loops after FANCM depletion. To test whether FANCM depletion impacts the XPF recruitment to telomeres, we quantified the colocalization of XPF and TRF2 by immunostaining. FANCM depletion significantly increased the colocalization events and exhibited larger XPF foci that were colocalized with TRF2 (Fig. 5a, b). FANCM depletion displayed a significant increase of telomeric R-loops (Fig. 5c), APBs (Fig. 5d, Supplementary Fig. 5c, d), and γH2AX foci at telomeres (Fig. 5e, f), indicating that undissolved telomeric R-loops are associated with DDR at ALT telomeres. To elucidate if accumulation of R-loops caused by FANCM deficiency contributes to DDR at ALT telomeres, we established a tetracycline-inducible system to overexpress RNase H1 in FANCM knockdown cells. Induction of RNase H1-mCherry expression by doxycycline attenuated the elevation of telomeric R-loops, APBs, γH2AX at telomeres, and top 5% telomere intensity caused by FANCM deficiency (Supplementary Fig. 6a–c), suggesting that FANCM-deficiency induced R-loops promote DDR at telomeres and ALT features.

Next, we performed the double-knockdown experiment for XPF and FANCM. Strikingly, XPF knockdown significantly suppressed the formation of γH2AX foci at telomeres in FANCM deficient cells (Fig. 5e, f) but did not decrease the level of APBs (Fig. 5d, Supplementary Fig. 5d). The S9.6 DRIP-qPCR showed an increase of telomeric R-loops after XPF depletion in both FANCM-proficient and FANCM-deficient cells, supporting that XPF is involved in telomeric R-loop processing (Fig. 5c). Moreover, the elevation of top 5% telomere intensity induced by FANCM deficiency was blocked by the loss of XPF (Fig. 5g). We conclude that XPF is required for the DDR induced by FANCM deficiency at ALT telomeres but is not needed for APB formation.

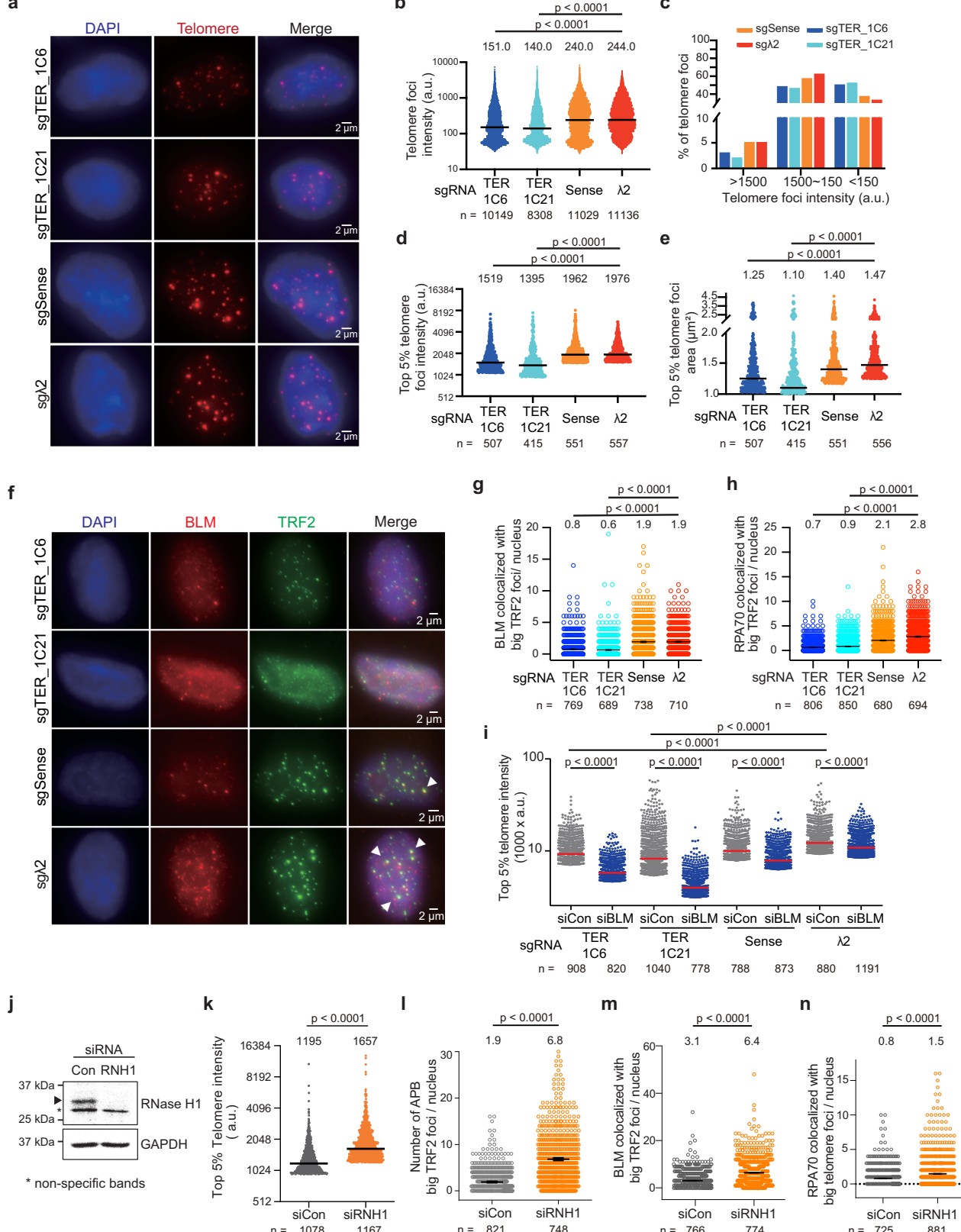

## CSB and SLX4 are dispensable for XPF-mediated DDR at telomeres

Observations of R-loop accumulation induced by deficiency of RNA processing factors revealed that transcription-coupled nucleotide excision repair factor (TC-NER) Cockayne syndrome group B (CSB), XPG, and XPF are all required for processing genomic R-loops into

DNA double-strand breaks (DSBs)[50]. We asked if CSB is required for DNA damage response at telomeres induced by FANCM deficiency. Unlike XPF knockdown, CSB depletion exhibited an elevation of γH2AX foci at telomeres (Supplementary Fig. 6d–f). Double knockdown of CSB and FANCM resulted in a robust increase of the formation of γH2AX foci at telomeres compared to FANCM single knockdown,

**Fig. 2 | TERRA R-loops promote telomere clustering. a** Telomere DNA FISH in TERRA knockdown cells (sgTER_1C6, sgTER_1C21) or control cells (sgSense, sgλ2). **b** Quantification of telomere intensity in U2OS cells expressing RCas9-sgRNAs. Each dot represents a telomere signal of each spot. Bars, medians. *n*, number of telomere foci. **c** Histogram showing the population of each telomere spot grouped by its intensity in cells expressing RCas9-sgRNAs. **d** Telomere clustering decreases in TERRA knockdown cells (sgTER_1C6, sgTER_1C21). Bars, medians of top 5% telomere intensity. n, number of telomere foci. **e** Quantification of the area of top 5% large telomere foci in cells expressing RCas9-sgRNAs. Bars, medians. *n*, number of telomere foci. **f** Immunostaining of BLM and TRF2 in U2OS cells expressing RCas9-sgRNAs. BLM foci at clustered telomeres decrease in TERRA knockdown cells. Arrowheads indicate BLM with big TRF2 foci. **g** Quantification of BLM foci at clustered telomeres determined by the colocalization of BLM with big TRF2 foci. *n*, cell number. **h** Quantification of the colocalization of RPA70 with big TRF2 foci. *n*, cell number. **i** Telomere clustering determined by top 5% telomere intensity decreases after BLM knockdown in cells expressing RCas9-sgRNAs. Bars, medians. *n*, number of telomere foci. **j** Western blot analysis shows the expression of RNase H1 after siRNA knockdown for 3 days. **k** Telomere clustering determined by top 5% telomere intensity increases after RNH1 knockdown. Bars, medians. n, number of telomere foci. **l** Quantification of the number of APB (PML-large TRF2 colocalized) foci after RNH1 knockdown. *n*, cell number. **m** Quantification of the number of BLM foci that were colocalized with large TRF2 after RNH1 knockdown. *n*, cell number. **n** Quantification of the number of RPA70 foci that were colocalized with large TRF2 after RNH1 knockdown. *n*, cell number. **b–e, i, k** Representative of three independent experiments. Other replicates show similar trends and are provided in the Source Data file. **g, h, l–n** Data of three independent experiments. Bars, mean ± SEM. **b, d, e, g, h, i, k–n** *P*-values by two-sided Mann-Whitney test. Mean or median values of each group shown on the top of figures.

indicating that CSB is not required for DSBs induced by FANCM deficiency at telomeres in ALT cells, and that is opposite to the observation in XPF-FANCM double knockdown. Thus, XPF-mediated DDR at ALT telomeres is not dependent on CSB.

SLX4 (SLX4 Structure-Specific Endonuclease Subunit) was shown to cooperate with XPF for interstrand DNA crosslink repair[35] and interact with telomeres[53]. We then tested if SLX4 is required for XPF-mediated DDR at ALT telomeres. Notably, double knockdown of SLX4 and FANCM exhibited an increase of γH2AX foci at telomeres (Supplementary Fig. 6d–f) compared to FANCM single knockdown, suggesting that SLX4 is dispensable for XPF-mediated DSBs at telomeres in ALT cells.

### XPF facilitates the loading of BRCA1 and RAD51 at telomeres in FANCM deficient cells

Next, we asked whether XPF contributes to homologous recombination (HR) to activate ALT activity. HR requires the presence of a 3′ overhang coated with the RAD51 recombinase to initiate the strand-invasion event[54], and BRCA1 contributes to the formation of 3′ overhang[55]. Upon FANCM depletion, there was a significant increase of colocalization of TRF2 with RAD51 and BRCA1 (Fig. 6a–c, Supplementary Fig. 7a), representing the events of the single-strand extruding for HR activity at telomeres. Remarkably, loss of XPF reduced the colocalization of TRF2 with BRCA1 and RAD51 in FANCM-deficient cells (Fig. 6a–c, Supplementary Fig. 7a), suggesting that XPF initiates a DNA break to promote the recruitment of BRCA1 and RAD51 for homologous recombination at ALT telomeres.

### XPF promotes break-induced telomere synthesis in ALT cells

One of the features of ALT cells is that telomeres are able to be synthesized in the G2/M phase of the cell cycle[7,8]. Loss of FANCM promotes non-S phase telomere synthesis[49] that can be induced by replication stress[51] or DSBs[7,24]. We noticed that telomere intensity was significantly increased upon FANCM depletion, but was dramatically hindered when XPF was depleted (Fig. 5g). However, XPF depletion did not decrease the number of APBs with large TRF2 foci (Fig. 5d, Supplementary Fig. 5d), which correspond to telomere clustering events in FANCM deficient cells. This observation led us to question whether the reduction of telomere intensity in XPF knockdown cells (Fig. 5g) was due to the alteration of telomere synthesis rather than telomere clustering. To monitor DNA synthesis at telomeres in the G2/M phase, U2OS cells were synchronized using CDK1 inhibitor, RO-3306, and labeled with 5-ethynyl-2′-deoxyuridine (EdU). The number of EdU foci at telomeres in non-S phase cells was significantly increased in FANCM knockdown cells, whereas it was significantly decreased in XPF knockdown cells (Fig. 6d, e, Supplementary Fig. 7b, c). Interestingly, XPF/FANCM double-knockdown recapitulated the phenotype of the reduction of EdU foci at telomeres in XPF single knockdown cells, indicating that XPF is required for non-S phase telomere synthesis induced by FANCM deficiency. As break-induced telomere synthesis

underlies alternative telomere maintenance and requires DNA polymerase δ (Pol δ)[7,25], we asked if XPF is involved in the recruitment of Pol δ at telomeres. Indeed, XPF depletion disrupted the colocalization of POLD3 (a Pol δ subunit) and TRF2 in FANCM proficient and deficient cells (Fig. 6f, Supplementary Fig. 7d), suggesting that XPF promotes break-induced telomere synthesis via driving DNA damage response to facilitate the recruitment of Pol δ.

As TERRA knockdown cells have reduced XPF recruitment to telomeres (Fig. 4c), we tested if telomere synthesis is altered in these cells. After synchronizing cells in the G2 phase, the number of EdU foci at telomeres was significantly lower in U2OS cells stably expressing sgTERRA-RCas9 in comparison to control cells (Fig. 6g, Supplementary Fig. 7e). To further validate the phenotype caused by downregulation of TERRA transcription, we performed transient TERRA depletion by viral transduction to express RCas9-sgRNAs. The results showed that cells transduced with RCas9-sgTERRA exhibited reduced colocalization of γH2AX foci with TRF2, telomere intensity, and the number of EdU foci at telomeres (Fig. 6h, Supplementary Fig. 8a–c). The data are consistent with the previous study[32] revealing that TERRA is responsible for break-induced telomere synthesis.

To confirm the role of XPF in break-induced telomere synthesis, another ALT-dependent cell line, WI38-VA, was transfected with siRNAs to silence XPF or FANCM. Consistently, the number of EdU foci colocalized with telomeres was significantly lower upon XPF depletion in WI38-VA cells (Fig. 6i). Collectively, these results demonstrate that the break-induced telomere synthesis is activated by the accumulation of TERRA and XPF at telomeres.

### XPF is required for telomere lengthening and cell proliferation in ALT cells

To elucidate the impact of XPF on telomere lengthening in ALT cells, we silenced XPF in U2OS cells for 8 weeks (Fig. 7a). Transfection of siRNAs was conducted once a week for three days, and cells were then recovered for four days after transfection. Strikingly, TRF analysis demonstrated that prolonged XPF depletion resulted in shorter telomere length compared to control cells (Fig. 7a). In addition, we observed that XPF depletion declined the growth rate much more in ALT cells (U2OS and WI38-VA) than non-ALT cells (HeLa) (Fig. 7b, Supplementary Fig. 8d). We conclude that XPF is crucial for telomere lengthening and cell proliferation in ALT cells.

Conventional chemotherapy drugs can induce DNA breaks to trigger cell death and usually are commonly used for the treatment of sarcoma, which tends to utilize ALT for the extension of their telomeres[10]. We surmised that the poor prognosis of ALT cancers[11] might be due to the resistance to conventional chemotherapy drugs, and wondered if XPF could be a drug target for ALT cancers. To test the drug sensitivity, ALT cells (U2OS and WI38-VA) and non-ALT cells (HeLa, HT-1080, and SK-LMS-1) were treated with conventional chemotherapy drugs and a ERCC1-XPF inhibitor NSC130813[56] (Fig. 7c, Supplementary Fig. 8e). Doxorubicin and etoposide both interact with

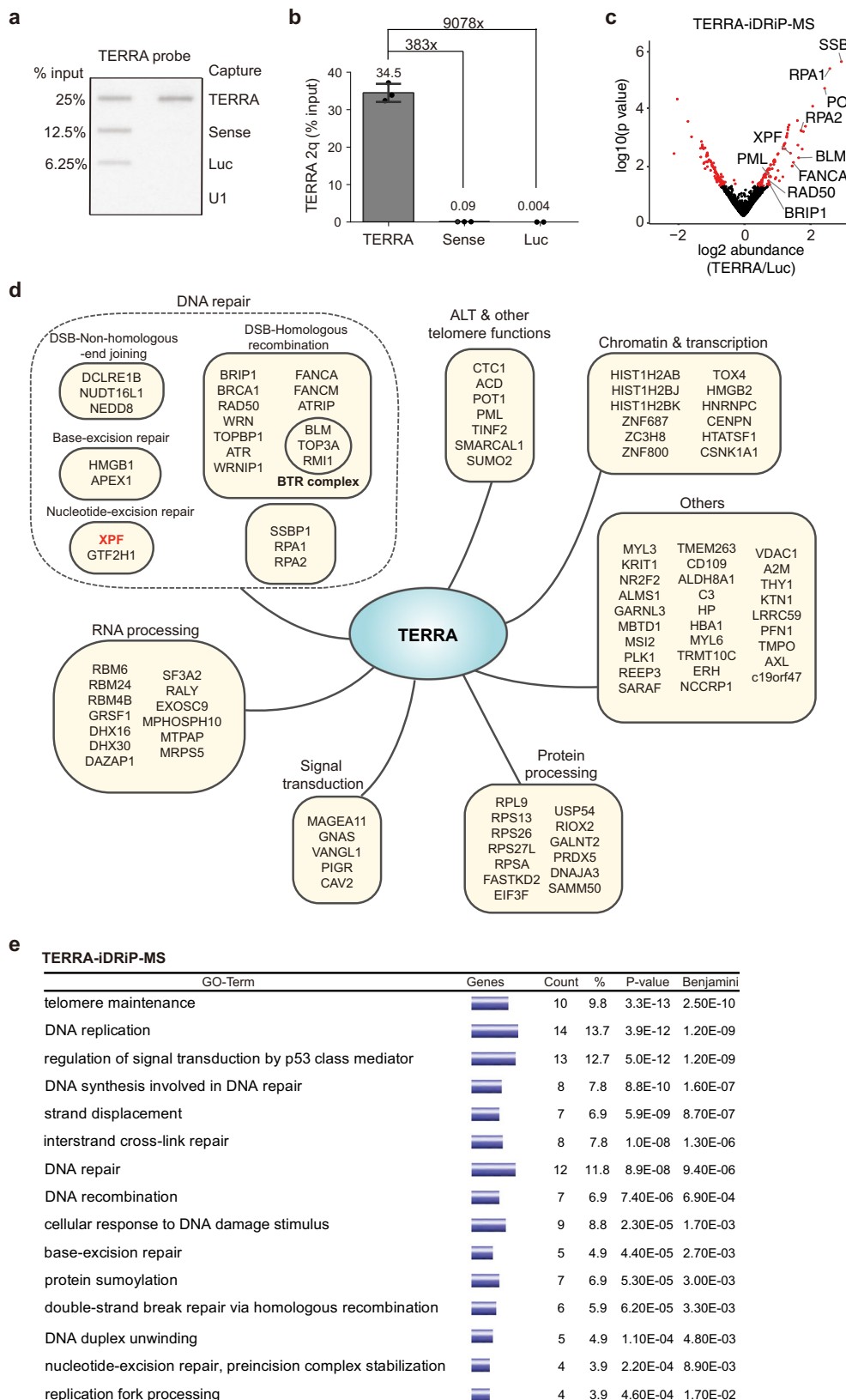

**Fig. 3 | TERRA interacts with DNA repair proteins in ALT cancer cells. a** RNA slot blotting shows the enrichment of TERRA RNA after iDRiP capture in U2OS cells. Two independent experiments show similar results. **b** Quantification of TERRA derived from chromosome 2q by qRT-PCR. Bars, mean ± SD. Representative of four independent experiments. **c** Volcano plot showing proteome abundance of TERRA iDRiP-MS normalized with luciferase iDRiP-MS. *P*-values calculated by Rank Product method. **d** TERRA interacting proteins are subclassified into functional groups. Data of TERRA-iDRiP-MS enriched proteins from four biological replicates. **e** Gene Ontology analysis of TERRA interacting proteins identified enriched pathways by DAVID bioinformatics resources. *P*-values by Fisher Exact test. Benjamini FDR was shown.

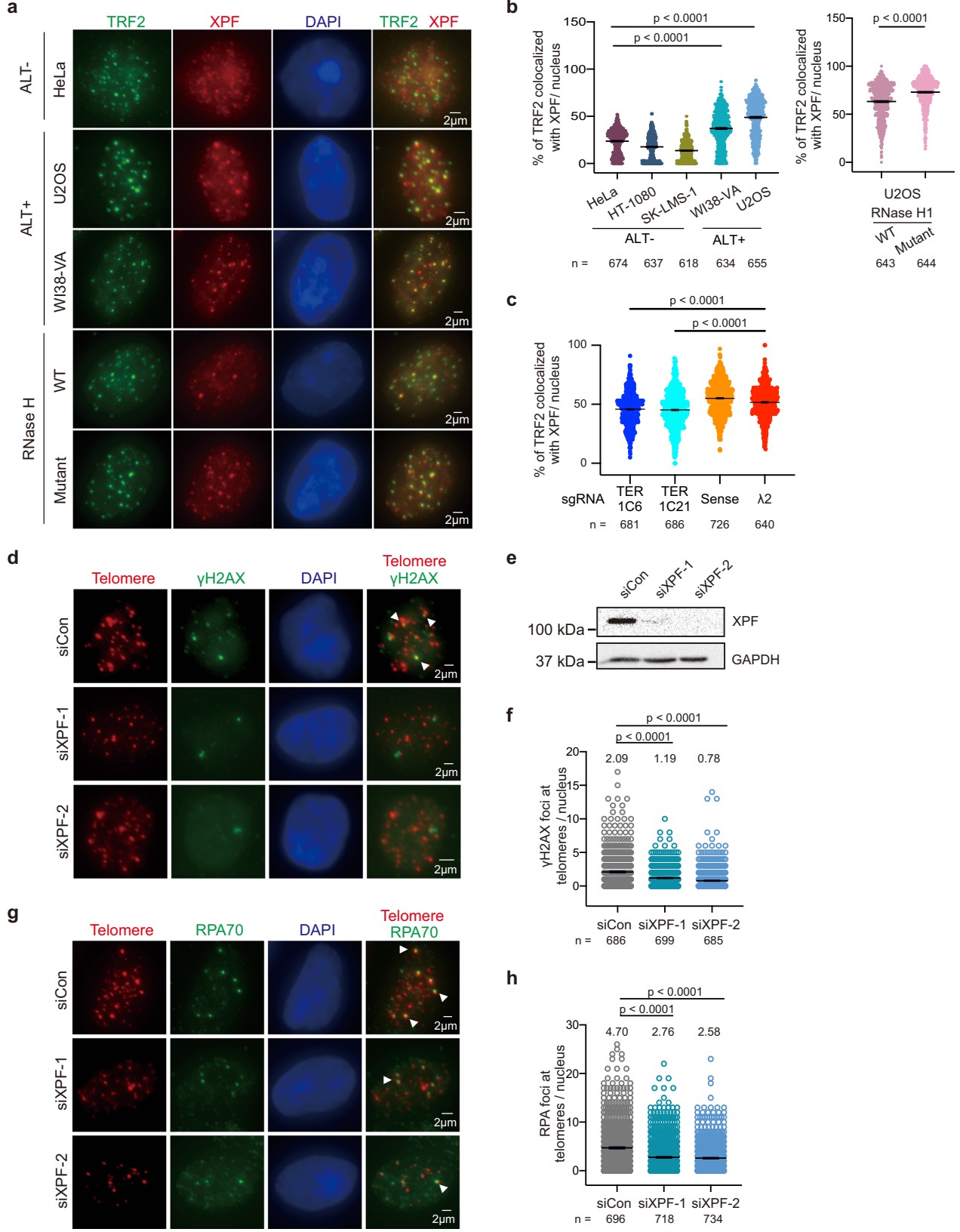

DNA and inhibit topoisomerase II to generate DSBs[57,58]. Cisplatin particularly reacts with the nucleophilic N7-sites of purine bases and causes DNA damage[59]. Our results indicated that U2OS cells were more resistant to doxorubicin, etoposide and cisplatin, in comparison to non-ALT cells (Fig. 7c, Supplementary Fig. 8e). WI38-VA cells were more resistant to doxorubicin and etoposide compared to non-ALT

cells. The treatment of ERCC1-XPF inhibitor NSC130813[56] (Fig. 7c) exhibited a higher toxicity in U2OS cells (ALT) than in HeLa cells (non-ALT). The effect of NSC130813 on XPF and ERCC1 was determined by immunostaining (Supplementary Fig. 8f) that showed a decrease of XPF-ERCC1 colocalization events. The protein levels of XPF and ERCC1 were also reduced (Supplementary Fig. 8g) in U2OS cells treated with

**Fig. 4 | XPF is recruited by TERRA R-loops to generate DNA damage response. a** Representative images of XPF and TRF2 staining in HeLa, WI38-VA, and U2OS cells. U2OS cells were expressing wildtype RNase H1 or catalytic-dead mutant RNase H1. **b** Quantification of the percentage of TRF2 colocalized with XPF per nucleus in various cell lines and in U2OS cells expressing WT and mutant RNase H1. Percentage of TRF2 co-localized with XPF = (XPF-TRF2 co-localization events / total TRF2 foci). Data of three independent experiments. **c** Quantification of the percentage of TRF2 colocalized with XPF per nucleus in U2OS cells expressing RCas9-sgRNAs. Data of three independent experiments. Other replicates show similar trends and are provided in the Source Data file. **d** Representative images of immuno-DNA FISH to detect the colocalization of γH2AX and telomeres in U2OS cells after 72 h transfection with control or XPF siRNAs. Arrowheads indicate colocalization events. **e** Western blot analysis for XPF in U2OS cells after siRNA knockdown. GAPDH, loading control. **f** Quantification of γH2AX foci at telomeres. Data of three independent experiments. **g** Representative images of immuno-DNA FISH to detect the co-localization of RPA70 and telomeres in U2OS cells after transfection with control or XPF siRNAs. Arrowheads indicate colocalization events. Data of three independent experiments. **h** Quantification of RPA70 foci at telomeres. Data of three independent experiments. **a–h** Mean or median values of each group shown on the top of figures. *P*-values by two-sided Mann-Whitney test. Bars, mean ± SEM. *n*, cell number.

NSC130813. When comparing more cell lines, ALT cells and non-ALT cells displayed similar sensitivity to NSC130813 (Supplementary Fig. 8e), indicating that ALT cells are not more resistant to NSC130813.

For the reason that XPF promotes break-induced telomere synthesis, loss of XPF can have an adverse effect in ALT cancer cells. In sum, our finding uncovers an important role of XPF in ALT cells that could be exploited for cancer therapy.

## Discussion

We addressed several questions related to how TERRA and XPF regulate the ALT pathway (Fig. 7d), and demonstrated that TERRA associated R-loops initiate the recruitment of DNA damage response proteins such as BLM, RPA, and XPF to telomeres. Suppressing TERRA levels using the RCas9 system in ALT cells declined ALT features including APBs, telomere clustering, telomere lengthening and replication (Figs. 1, 2, 6). We found that NER factors such as XPF and general transcription factor II H (GTF2H1) were enriched in TERRA-iDRiP-MS (Fig. 3d). Importantly, inhibiting the formation of TERRA R-loops by TERRA depletion or overexpression of RNase H1 reduced XPF localization at telomeres (Fig. 4), while accumulation of TERRA R-loops caused by FANCM deficiency increased XPF recruitment to telomeres (Fig. 5a, b), suggesting that telomeric R-loops promote the recruitment of XPF to telomeres. Moreover, depletion of XPF suppressed the formation of γH2AX, the recruitment of BRCA1 and RAD51, and telomere synthesis at ALT telomeres in FANCM deficient cells (Figs. 5, 6). Prolonged XPF depletion disrupts telomere lengthening in U2OS cells (Fig. 7a). Altogether, our results show that XPF activates DDR induced by TERRA R-loops and promotes break-induced telomere replication for Alternative Lengthening of Telomeres (Fig. 7d).

TERRA-iDRiP-MS from ALT cells uncovered that TERRA interacts with a large number of proteins functioning in several DNA repair pathways, including homologous recombination, NER, non-homologous end joining, interstrand crosslink repair, base-excision repair (Fig. 3), suggesting that TERRA interplays with these proteins to regulate ALT. Comparing TERRA-iDRiP-MS in ES cells[43] and ALT human cancer cells, the common enriched proteins are involved in telomere capping and the ALT pathway (Supplementary Fig. 3d). Remarkably, more DNA repair proteins are enriched in ALT cells (Fig. 3d), whereas more chromatin remodelers are enriched in mouse ES cells[43], implying that TERRA plays different roles depending on the cellular circumstance. For instance, ATRX, a TERRA interacting protein[43], which suppresses the formation of TERRA R-loops[60], is usually lost or mutated in ALT cells[61]. This could explain why ALT cells tend to accumulate TERRA R-loops.

DSB-induced homologous recombination occurs at APBs to promote telomere synthesis in ALT cells[4,27]. APBs are observed as large foci composed of multiple telomeres from different chromosomes in association with PML[12,14,15,62]. However, how APBs are formed specifically in ALT cells is unclear. A recent study has shown that inducing telomere clustering by polysumoylation combined with overexpressing BLM artificially generates APB-like condensates and elicits the ALT activity[46], revealing that telomere clustering and BLM are required for the onset of the ALT pathway. We demonstrate that depletion TERRA suppresses APB formation, telomere clustering, and BLM recruitment to ALT telomeres (Fig. 2, Supplementary Fig. 2), implying that TERRA plays an important role in the initiation of the ALT mechanism. It seems that TERRA could serve as a guide or a scaffold to tether proteins such as BLM or RPA and telomeric DNA together to promote APB formation. Moreover, evidence has shown that transient inhibition of TERRA transcription by tethering transcription repressors to subtelomeric regions suppresses DNA damage at telomeres and alleviates ALT activity[32] such as APB formation and break-induced telomere synthesis. In agreement with the previous study, TERRA depletion by RCas9-sgRNA also displayed reduced DDR at ALT telomeres and telomere synthesis (Fig. 6g, h, Supplementary Fig. 8a–c). These results support that TERRA transcripts promote DNA damage signals to initiate ALT.

Consistent with the previous study showing that FANCM resolves TERRA R-loops in vitro[49], we observed that depletion of FANCM leads to accumulation of TERRA R-loops, APBs, and robust DDR at ALT telomeres (Fig. 5). Co-depletion of XPF and FANCM abolished γH2AX foci at telomeres, indicating that XPF is required for the DDR induced by FANCM deficiency. Unexpectedly, we observed that XPF depletion declined telomere intensity and telomere synthesis, but not the formation of large TRF2 foci within APBs (Fig. 5). These results indicate that newly synthesized DNA fragments containing telomeric repeat sequences do not directly correspond to the abundance of TRF2 at ALT telomeres. On the other hand, TERRA depletion reduced APBs with large TRF2 foci and telomere intensity, suggesting that TERRA regulates the early events of ALT including the formation of APBs and telomere clustering. APBs are usually associated with DNA damage response at ALT telomeres. To our surprise, loss of XPF did not alter APB formation but robustly reduced the formation of γH2AX and telomere synthesis (Figs. 4–6). Therefore, it is possible that DNA damage response is independent of APB formation and the two stages of ALT can be separated: First, TERRA R-loops induce APB formation; second, DNA damage response mediated by XPF induces HR and telomere synthesis (Fig. 7d). Since only a fraction of cells showed DNA damage responses at ALT telomeres by immunostaining, those events could be very dynamic and occur transiently. HR intermediates should be resolved when break-induced telomere synthesis is completed.

Emerging evidence indicates that R-loop formation is induced by prolonged stalled RNA polymerase II (Pol II) and could result in collisions with replication forks[63–65]. Pol II-blocking DNA lesions are removed by transcription-coupled repair (TC-NER)[66,67], a specialized sub-pathway of nucleotide excision repair[68–70] that depends on CSA and CSB proteins. CSB is associated with Pol II at transcriptional blocking sites and is responsible for the attraction of NER factors[71,72]. Ablation of CSB, XPF, and XPG reduce DNA damage response induced by RNA/DNA helicase Aquarius (AQR) deficiency[50], indicating that CSB, XPF, and XPG are all required for processing genomic R-loops into DNA double-strand breaks (DSBs). Similarly, we observed that XPF and XPG are required for DNA damage response at ALT telomeres (Fig. 4, Supplementary Fig. 4). However, CSB is not required for DSBs at ALT telomeres. Depletion of CSB has an additional effect on the formation of γH2AX at telomeres in FANCM-deficient cells, whereas XPF

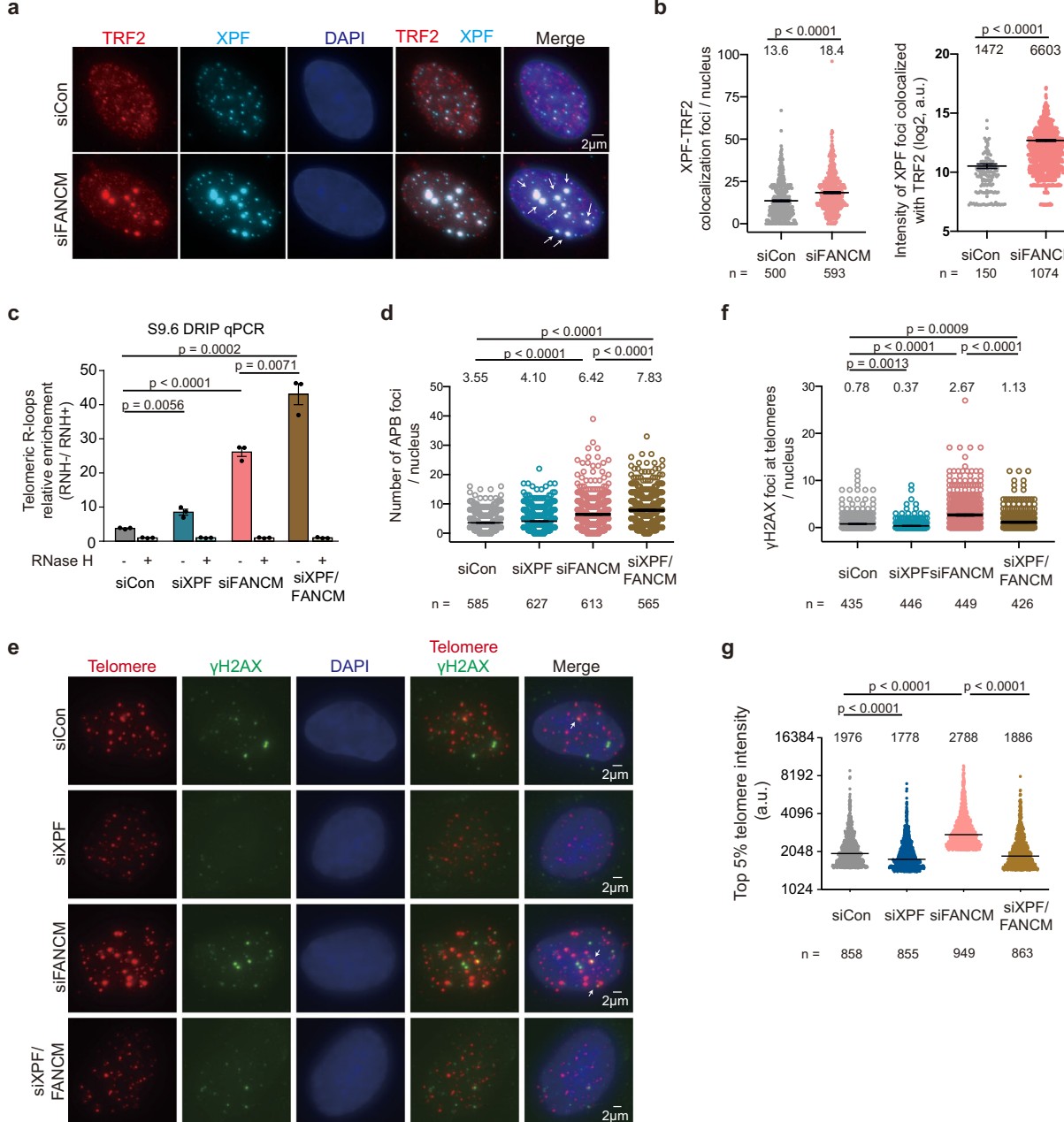

**Fig. 5 | DDR at ALT telomeres induced by FANCM deficiency is mediated by XPF. a** Representative images of TRF2 and XPF co-immunostaining in U2OS cells transfected with FANCM siRNAs for 3 days. Arrows indicate the colocalization events. **b** Quantification of colocalization events in (**a**). The number of XPF-TRF2 colocalization per nucleus (left). The intensity of XPF foci that are colocalized with TRF2 (right). *P*-values by two-sided Mann-Whitney test. Bars, mean ± SEM. *n*, cell number (left) or number of XPF foci (right). Data of three independent experiments. **c** DRIP-qPCR to detect telomeric R-loops in U2OS cells after transfection with control, XPF, or FANCM siRNAs for 3 days. Relative R-loop levels were normalized to the same amount of genomic DNA treated with RNase H prior to DRIP. *P*-values by two tailed Student's *t*-test. Bars, mean ± SD. Representative of three independent experiments. Other replicates show similar trends and are provided in the Source Data file. **d** Quantification of APBs in U2OS cells transfected with siRNAs. APB foci were determined by large TRF2 foci with PML staining shown in Supplementary Fig. 5d. *n*, cell number. Bars, mean ± SEM. *P*-values by two-sided Mann-Whitney test. Data of three independent experiments. **e** Representative images of immuno-DNA FISH to detect the co-localization of γH2AX and telomeres in U2OS cells after transfection with control, XPF, or FANCM siRNAs for 3 days. Arrows indicate the colocalization events. **f** Quantification of γH2AX foci at telomeres in (**e**). *n*, cell number. Bars, mean ± SEM. *P*-values by two-sided Mann-Whitney test. Data of two independent experiments. **g** Quantification of top 5% telomere intensity foci in (**e**). *n*, cell number. Bars, medians. *P*-values by two-sided Mann-Whitney test. Representative of three independent experiments. Other replicates show similar trends and are provided in the Source Data file. **a**–**g** Mean or median values of each group shown on the top of figures.

knockdown suppresses DNA damage response at telomeres induced by FANCM-deficiency. Correspondingly, CSB promotes DNA repair of oxidative stress-induced telomeric DSBs[73]. These data hint that CSB functions in DNA repair at telomeres, but DSBs induced by FANCM deficiency at telomeres are not mediated by CSB (Supplementary Fig. 6). As TERRA can act in trans[44,74] and bind to chromatin, it is likely

that TERRA invading double-stranded DNA to form an R-loop structure can be uncoupled with transcription activity. In support of this idea, RAD51 is capable of driving TERRA R-loop formation in trans[74], and tethering RNA by dCas9 to the DSB promotes HR by forming R-loops[75]. Thus, the formation of TERRA R-loops in trans could be important for the activation of HR to promote ALT.

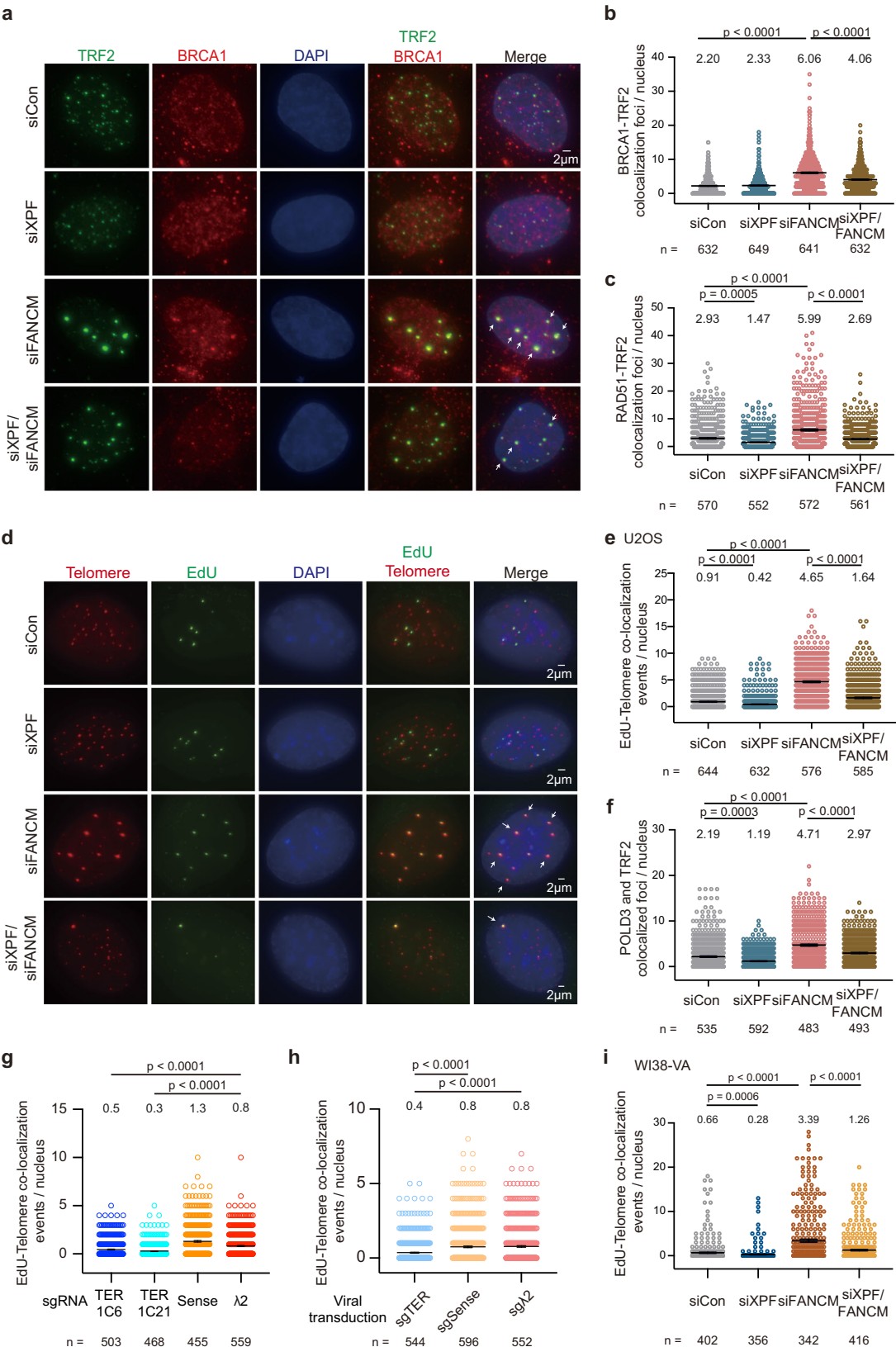

Elongation of ALT telomeres involves a replication stress-associated[24] and break-induced replication (BIR) process[7,8]. Artificially induction of telomeric DSBs by overexpressing TRF1-FokI triggers robust DNA synthesis that requires Pol δ[7]. Accumulation of TERRA R-loops in FANCM-deficient cells[52] or stabilization of G4 structures by G4 ligands[24] induce replication stress and DSBs that elicit telomere DNA synthesis. Therefore, it is believed that substantial DNA damage must be accumulated in order to activate the DNA repair mechanism to promote DNA synthesis at ALT telomeres[49]. However, the prolonged DNA damage and unresolved replication stress at ALT telomeres caused by FANCM deficiency are toxic to ALT cells[76]. We observed that the depletion of XPF displayed a significant reduction of the

**Fig. 6 | XPF promotes HR and break-induced telomere synthesis.**
**a** Representative images of TRF2 and BRCA1 immunostaining in U2OS cells transfected with control, XPF, FANCM, or XPF/FANCM siRNAs for 3 days.
**b** Quantification of colocalization events of BRCA1 and TRF2 in (**a**). **c** Quantification of colocalization events of RAD51 and TRF2 in U2OS cells transfected with control, XPF, FANCM, or XPF/FANCM siRNAs for 3 days. **d** Representative images of EdU-DNA-FISH to detect telomere synthesis in siRNA transfected U2OS cells after the treatment of RO-3306. Arrows indicate the colocalization events. **e** Quantification of EdU and telomere colocalization in non-S phase cells in (**d**). **f** Quantification of POLD3 and TRF2 colocalization events in U2OS cells transfected with siRNAs after

the treatment of RO-3306. **g** Quantification of EdU and telomere colocalization events in stable cell lines expressing RCas9-sgRNAs. TERRA knockdown cells (sgTER_1C6, sgTER_1C21) or control cells (sgSense, sgλ2). **h** Quantification of EdU and telomere colocalization events in U2OS cells with transient TERRA knockdown by viral transduction with RCas9-sgRNA plasmids. **i** Quantification of EdU and telomere colocalization events in WI38-VA cells transfected with control, XPF, FANCM, or XPF/FANCM siRNAs. **a**–**i** Data of three independent experiments. *n*, cell number. Bars, mean ± SEM. Mean values of each group shown on the top of figures. *P* values by two-sided Mann-Whitney test.

recruitment of BRCA1, RAD51, and EdU foci at ALT telomeres in FANCM-deficient cells (Fig. 6), suggesting that XPF contributes to homologous recombination and telomere synthesis induced by the accumulation of TERRA R-loops. On the other hand, XPF-induced DDR at telomeres is independent of SLX4 (Supplementary Fig. 6). Depletion of SLX4 in FANCM deficient cells results in an increase of γH2AX at telomeres, whereas depletion of XPF shows less DDR (Fig. 5e, f) but slightly increases APB foci (Fig. 5d). Thus, SLX4 is not involved in XPF-mediated DDR but more likely might cooperate with XPF for disruption of APB formation induced by replication stress[77]. It is possible that XPF has different roles during ALT. Initially, XPF is recruited by R-loops to generate DNA double-strand breaks at ALT telomeres to induce HR and trigger break-induced telomere synthesis. Later, XPF cooperates with SLX4 in the resolution of HR intermediates after telomere synthesis.

In FANCM-proficient cells, XPF depletion suppresses the formation of γH2AX foci, Pol δ recruitment, and EdU foci at telomeres (Figs. 5, 6), indicating that XPF is required for the maintenance of the basal levels of DDR at telomeres to trigger break-induced telomere synthesis. Moreover, prolonged XPF deficiency decreases telomere length in ALT cells (Fig. 7a), and the ablation of XPF hinders cell growth more severely in ALT than in non-ALT cells (Fig. 7b). Evidence has shown that exogenously expressing XPF in telomerase-positive cells inhibits telomere lengthening independent of XPF nuclease activity[78]. It seems that the effect of XPF on telomere maintenance might be very different between ALT and non-ALT cells. Telomeres in non-ALT cells prefer telomere fusion when telomeres are uncapped[39], but the ALT telomeres tend to utilize HR and break-induced replication induced by DNA damage response that is mediated by XPF. These data reveal a critical role of XPF specifically in ALT cells. XPF is responsible for DDR at telomeres. Silencing XPF blocks DDR caused by loss of FANCM, which is capable of unwinding D-loops[79] and R-loops in vitro[49]; therefore, these results imply that XPF contributes to DNA breaks induced by persistent DNA secondary structures such as R-loop, D-loop, as well as G-quadruplex in response to telomeric replication stress.

Our results (Fig. 7c) demonstrate that ALT cells are resistant to conventional chemotherapy drugs including doxorubicin and etoposide. The underlying mechanisms of these observations remain unclear and require more investigations. ALT and non-ALT cells showed a similar drug sensitivity to NSC130813. Further studies are needed for the examination of the specificity of NSC130813 and the development of XPF inhibitors.

As the proteomic data revealed TERRA associated proteins in various pathways in ALT cancer cells, it provides extensive resources for the exploration of the ALT mechanism.

## Methods
### Cell culture
U2OS, WI38-VA, HeLa, HT-1080, and SK-LMS-1 cells were cultured in DMEM supplemented with fetal bovine serum, L-glutamine, penicillin/streptomycin, glucose, and sodium pyruvate. Cells were mycoplasma free (routinely tested for mycoplasma) and maintained in a humidified incubator at 37 °C with 5% $CO_2$ and passaged every 2-3 days with Trypsin-EDTA (0.25%). U2OS cell lines with overexpressing the wild-

type RNase H1 (pICE-RNaseH1-WT-NLS-mCherry, Addgene #60365) and mutant RNase H1 (enzyme dead, pICE-RNaseH1-D10R-E48R-NLS-mCherry, Addgene #60367) were generated as described[52]. pICE-NLS-mCherry plasmid (vector alone) without RNaseH1-WT was generated by removing RNaseH1-WT from pICE-RNaseH1-WT-NLS-mCherry. For the construction of tetracycline-inducible RNase H1, RNase H1-WT-mCherry DNA sequence was amplified from pICE-RNaseH1-WT-NLS-mCherry by PCR and introduced into PB-TRE plasmid (Addgene, #63800) in which dCas9-VPR was replaced by RNaseH1-WT-NLS-mCherry. U2OS cells were transfected with PB-TRE-RNase H1-mCherry and selected by 200 μg/ml hygromycin B for one week. The survived cell colonies were picked and maintained in 50 μg/ml hygromycin B containing medium. Positive clones were further confirmed by DNA sequencing and fluorescent signals via live imaging. 50 ng/mL of doxycycline was added into cells for 24 h to induce RNase H1 expression.

### iDRiP
The detailed method was described previously[80]. Briefly, U2OS cells ($3.75 \times 10^8$ cells) were irradiated with UV light at 400 mJ energy (Stratagene 2400) in a minimal amount of cold PBS, cells were treated with CSKT-0.5% (10 mM PIPES, pH 6.8, 100 mM NaCl, 3 mM $MgCl_2$, 0.3 M sucrose, 0.5% Triton X-100, 1 mM PMSF) for 10 min at 4 °C. Snap freezing cells were stored at −80 °C. UV-crosslinked cells were treated with 8 ml of DNase I solution (50 mM Tris pH 7.5, 0.5% Nonidet-P 40, 0.1% sodium lauroyl sarcosine, 1x protease inhibitors, SuperaseIn, 600U DNase I) at 37 °C for 20 min to solubilize the chromatin. The samples were further lysed in 1% sodium lauroyl sarcosinate, 0.1% sodium deoxycholate, 0.5 M lithium chloride, 20 mM EDTA, and 20 mM EGTA and incubated at 37 °C for 5 min. The lysates were spun at the highest speed and the supernatants were collected. The pellets were resuspended in lysis buffer (50 mM Tris pH 7.5, 0.5 M LiCl, 1% Nonidet-P 40, 1% sodium lauroyl sarcosine, 0.1% sodium lauroyl sarcosine, 20 mM EDTA, 20 mM EGTA), incubated on ice for 10 min, heated to 65 °C for 5 min, immediately spun at room temperature for 1 min, and stored on ice. The supernatants were collected and combined with the previous supernatants (the total volume was around 11–12 ml). The combined supernatants (3 ml for each probe capture) were precleaned by incubation with MyOne streptavidin C1 beads (ThermoFisher). To conjugate DNA probes to beads, beads were incubated with probes (0.5 nmol/ml of bead) in 1X binding buffer (5 mM Tri-HCl (pH7.5), 0.5 mM EDTA, 1 M NaCl) at room temperature for 20 min, washed with 1X binding buffer twice, and then resuspended in lysis buffer. Precleaned lysates and probes-conjugated beads were preheated to 65 °C, mixed, and incubated at 65 °C for 15 min (100 μl beads for 1 ml of lysate). Followed by slowly reducing the temperature to 37 °C, lysates were incubated at 37 °C for one hour. The beads were washed three times in Wash Buffer 1 (50 mM Tris, pH 7.5, 0.3 M LiCl, 1% SDS, 0.5% Nonidet-P 40, 1 mM DTT, 1 mM PMSF, 1X protease inhibitors) at 37 °C followed by treatment with 20 U of Turbo DNase I in DNase I digestion buffer (50 mM Tris pH 7.5, 0.5% Nonidet-P 40, 0.1% sodium lauroyl sarcosine) with the addition of 0.2 M LiCl, protease inhibitors (Merck, Cat# 4693132001), and superaseIn (ThermoFisher) at 37 °C for 10 min. Then, beads were washed two more times at 37 °C in the Wash

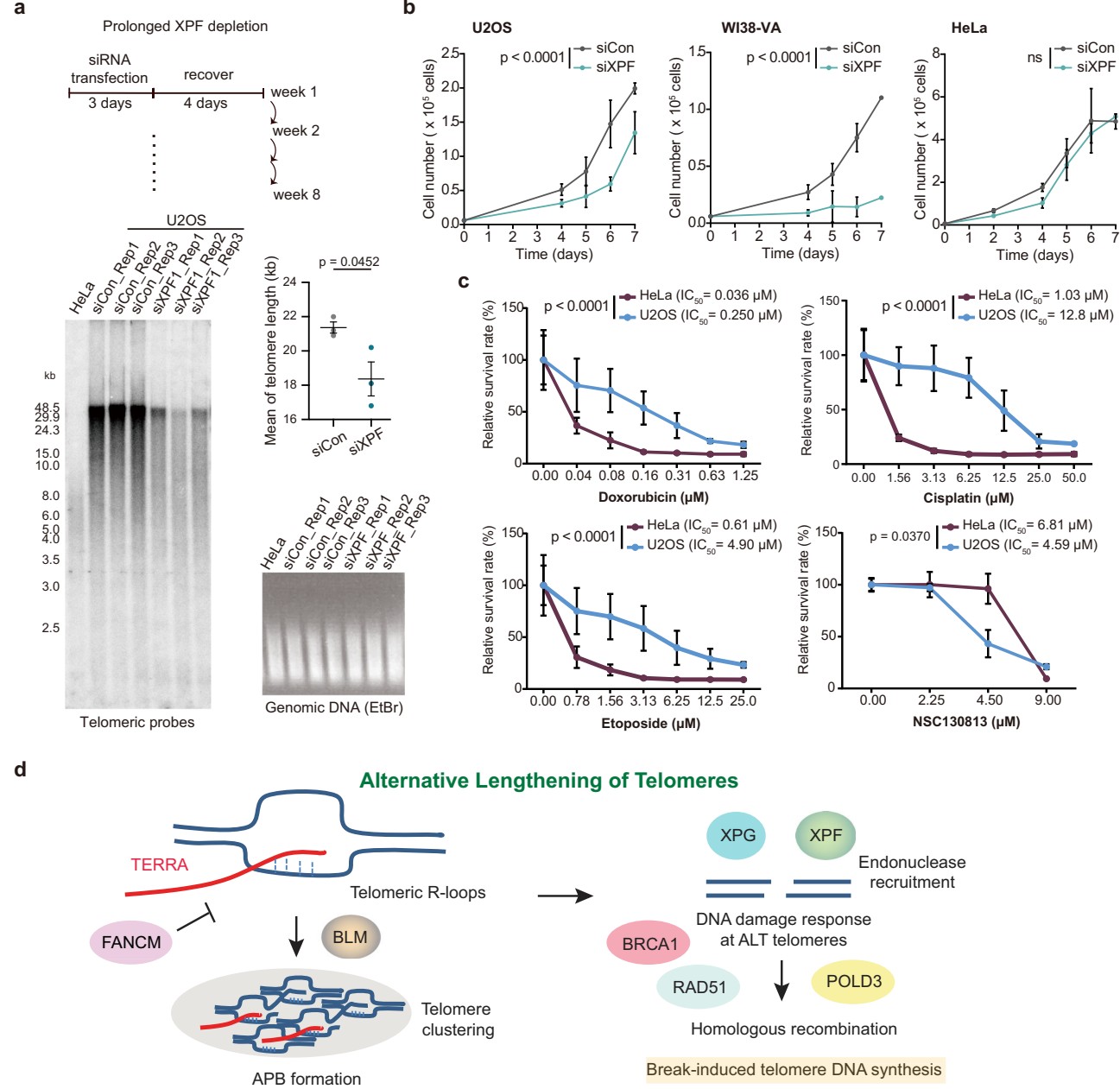

**Fig. 7 | Inhibition of XPF suppresses cell proliferation in ALT cells. a** Telomere length measured by TRF analysis in cells after prolonged XPF-depletion for 8 weeks in U2OS cells. The same amount of DNA as TRF assay was loaded in the EtBr gel. The mean of telomere length was quantified (top-right) from three independent knockdown experiments. *P*-values by two-tailed Student's *t*-test. Bars, mean ± SEM. **b** Cell proliferation assay after XPF knockdown in U2OS, HeLa, and WI38-VA cells. Bars, mean ± SD. *P*-values by two-way ANOVA. ns, no significance. Representative of four independent experiments. Other replicates show similar trends and are provided in the Source Data file. **c** Cell survival rates after the 3-day treatment of XPF-ERCC1 inhibitor (NSC130813), doxorubicin, etoposide, and cisplatin in ALT + (U2OS) and ALT- (HeLa) cells. *P*-values by two-way ANOVA. Bars, mean ± SD. At least three independent experiments were averaged. **d** Model of TERRA and XPF promoting break-induced telomere synthesis in ALT cells. TERRA promotes APB and R-loop formation at telomeres. XPF and XPG are recruited by TERRA R-loops to generate DNA double-strand breaks at ALT telomeres, thus facilitating HR and strand invasion to initiate break-induced telomere synthesis.

Buffer 1. The lysates were further washed at 37 °C for 5 min in Wash Buffer 2 (1% SDS, 1 mM DTT, 5 mM EDTA, 150 mM NaCl, 1 mM PMSF). Proteins were eluted in an elution buffer (10 mM Tris, pH 7.5, 1 mM EDTA, 0.05% Triton X-100) at 70 °C for 5 min. DNA probes for iDRiP were ordered from Integrated DNA Technologies and labeled with 3′ biotin-TEG.

### Quantitative mass spectrometry
**TMT labeling and peptide fractionation.** Similar to previously described[81], proteins enriched after iDRiP capture were suspended in 8 M urea/50 mM HEPES (pH 8), disulfide bonds were reduced, free thiols were alkylated with iodoacetamide; protein samples were transferred to the Amicon Ultra-0.5 centrifugal filters (10 kDa, Millipore, Burlington, MA, USA) and spun at 13,200 × g for 20 min. Buffer exchange was performed in four successive washes with 8 M urea in 50 mM HEPES (pH 8). Afterward, samples were digested overnight at 37 °C using LysC protease and trypsin at an enzyme-to-substrate ratio of 1:50 (w/w), and peptides were desalted using C$_{18}$ Stage Tips. Eluates were dried by vacuum centrifugation. Desalted peptides were reconstituted in 50 mM HEPES, and TMT10-plex reagents (Thermo Fisher) were added from stocks dissolved in 100% anhydrous ACN. The peptide-TMT mixture was incubated for 1 h at room temperature with

frequent mixing, and the labeling reaction was stopped by the addition of 5% hydroxylamine to a final concentration of 0.2% and incubation for 15 min at room temperature. Labeled peptide mixtures were pooled into 10-plexed samples, which was fractionated into 8 fractions by using a high pH reversed-phase peptide fractionation kit (Pierce). **LC-MS/MS measurements** - Each fraction was injected for nano-liquid chromatography-nano electrospray ionization-tandem mass spectrometry (nanoLC-nanoESI-MS/MS) analysis. NanoLC-nanoESI-MS/MS analysis was performed on an EASY-nLCTM 1200 system connected to a Thermo Scientific Orbitrap Fusion Lumos Tribrid Mass Spectrometer (Thermo Fisher Scientific, Bremen, Germany). Peptide mixtures were loaded onto a 75 µm ID, 25 cm length PepMap C18 column (Thermo Fisher Scientific) packed with 2 µm particles with a pore with of 100 Å and were separated using a segmented gradient in 120 min from 5% to 45% solvent B (80% acetonitrile with 0.1% formic acid) at a flow rate of 300 nl/min. Solvent A was 0.1% formic acid in water. The mass spectrometer was operated in the data-dependent mode. Briefly, survey scans of peptide precursors from 350 to 1600 $m/z$ were performed at 120 K resolution with a $2 \times 10^5$ ion count target. Peptide precursors with charge state 2–7 were sampled for MS2. Tandem MS was performed by isolation window at 0.7 Da with the quadrupole, HCD fragmentation with a normalized collision energy of 35, and MS2 spectra were acquired at 50 K resolution using an AGC target value of $1 \times 10^5$ and the max injection time was 100 ms. **Database searching** - Raw data files were processed using Proteome Discoverer v2.3 (Thermo Scientific) and the tandem MS data were then searched using SEQUEST algorithms against a human UniProt (Swiss-Prot only) database (released June 2019) with common contaminant proteins. The search parameters included trypsin as the protease with a maximum of 2 missed cleavages allowed; oxidation of methionine and deamidation of asparagine and glutamine were set as a dynamic modification while static modifications included carbamidomethyl (alkylation) at cysteine and TMT as a static modification of lysine residues and peptides' N-termini (+229.16293 Da). Precursor mass tolerance was set to 10 ppm and fragment mass tolerance was set to 0.02 Da. The false discovery rate (FDR) was calculated by carrying out decoy database searches and peptides scoring better than 1% FDR score cut-off were considered for further analysis. Reporter ions for TMT labeled peptides were quantified using the Reporter Ions Quantifier Node in Proteome Discoverer and peak integration tolerance was set at 20 ppm by considering most confident centroid peaks. **Statistical analysis** - Four biological replicates of TERRA-iDRiP were subjected to two different MS runs, and each run contained two biological replicates. The log2 abundances of two biological replicates in the same run were normalized with Luc control and averaged. *P*-values were calculated using the Rank Product method[82].

## RT-qPCR

Total RNA was isolated using TRIzol (ThermoFisher) and reverse-transcribed with random primers using Superscript IV reverse transcriptase (ThermoFisher). RT-qPCR was performed using iQ SYBR Green Supermix (Bio-Rad). The percentage of enrichment was compared to the input. A list of primers is provided in Supplementary Data 1.

## Western blot analysis

Cells were lysed in 2X sample buffer on ice for 10 min. Proteins were denatured at 95 °C for 5 min and stored at −80 °C. Western blotting was performed using a standard protocol, and antibodies against XPF (Thermo Fisher Scientific, cat#MA5-12060, dilution 1:1,000), FANCM (Merck, cat#MABC545, dilution 1:1,000), γH2AX (Cell Signaling, cat#9718 T, dilution 1:1,000), BRCA1 (Abcam, cat#ab16780, dilution 1:1,000), SLX4 (Bethyl Laboratories, cat#A302-269A, dilution 1:1,000), CSB (Bethyl Laboratories, cat#A301-345A, dilution 1:1,000), BLM (Bethyl Laboratories, cat#A300-110A, dilution 1:4,000), mCherry

(GeneTex, cat#GTX128508, dilution 1:5,000), tubulin (Santa Cruz, cat#sc-134239, dilution 1:5,000), RNase H1 (GeneTex, cat#GTX117624, dilution 1:1,000), ERCC1 (SantaCruz, cat#sc-17809, dilution 1:1,000) and GAPDH (Cell Signaling, cat#2118, dilution 1:5,000) were used. Antibody information is provided in Supplementary Data 1.

## Northern blotting and RNA slot blotting

Oligo probes were end-labeled using T4 polynucleotide kinase (New England Biolabs, cat#M0201). Total RNA was extracted using TRIzol followed by acid phenol extraction. Hybridization was carried out at 42 °C overnight using ULTRAhyb-Oligo hybridization buffer (ThermoFisher) or Church buffer. Probe information is provided in Supplementary Data 1.

## Immuno-RNA FISH

Cells grown on coverslips were washed with cold 1X PBS and treated with CSKT (10 mM PIPES, pH 6.8, 100 mM NaCl, 3 mM MgCl₂, 0.3 M sucrose, 0.5% Triton X-100, adjust to pH 6.8) for 10 min on ice. Cells were fixed in 4% paraformaldehyde at RT and stored at 70% EtOH at −20 °C. After washing with cold PBS, cells were incubated with blocking solution (1% BSA/PBS with 1 mM EDTA and 0.8 U/µl of RNase inhibitor) at 4 °C for 1 h. Cells were then incubated with primary antibodies in blocking solution at 4 °C overnight, and washed with 0.2% Tween20/PBS three times at 4 °C. Antibodies were used against TRF2 (Novus, NB110-57130, dilution 1:200) and PML (Santa Cruz, sc-966, dilution 1:100). Followed by incubation of secondary antibodies (Thermo Fisher Scientific, Alexa Fluor 555 goat anti-rabbit IgG or Alexa Fluor 555 goat anti-mouse IgG, dilution 1:500) in blocking solution at 4 °C for 2 h, cells were washed with PBS three times and fixed in 2% paraformaldehyde for 10 min. RNA FISH was then performed after immunostaining. TERRA oligo probes ((TAACCC)₇-Alexa-647-3′) for RNA-FISH were mixed at the final concentration of 0.5 pmol/µl in hybridization solution (50% formamide, 2 × SSC, 2 mg/ml BSA, 10% Dextran Sulfate-500 K). Hybridization was performed at 42 °C overnight for RNA FISH. Cells were washed with 2 × SSC/50% formamide for 5 min three times at 44 °C and then washed with 2 × SSC for 5 min twice at 44 °C. Images were captured using Olympus IX83 inverted microscopy with various Z-sections and then were compiled into 3D images to calculate APB and TERRA-associated APB foci. APB foci were determined by the extensive TRF2 staining (> top 5% of all TRF2 foci in the control cells, defined as large TRF2 foci) with PML staining. The antibody information is provided in Supplementary Data 1.

## Immuno-DNA FISH

For immuno-DNA FISH, telomere PNA (peptide nucleic acid) FISH was performed prior to immunostaining. Cells were dehydrated subsequently in 70, 80, 90, and 100% EtOH for 2 min each, and treated with RNase A (400 µg/ml) in 1 x PBS at 37 °C for 20 min. Coverslips were washed three times with 1x PBS for 5 min. Cells were dehydrated as previously described and slides were heated at 85 °C on the heating block. PNA probes (TelG-Cy3 or TelC-Alexa488) for telomeric DNA were diluted in PNA hybridization buffer (20 mM Tris-HCl, 70% formamide, 0.1 mg/ml salmon sperm DNA) at the final concentration of 50 nM and denatured at 85 °C for 5 min. Cells were incubated with PNA probes at 85 °C for 3 min. Slides on the heating block were removed from 85 °C to room temperature and waited for the temperature of the heating block to slowly decrease in the dark. Cells were washed twice with 0.2% Tween 20 in 1X PBS at 57 °C for 5 min and once at RT for 5 min. The immunostaining was performed as previously described. Antibody information is provided in Supplementary Data 1.

## Co-immunostaining

Cells were prepared as immune-RNA FISH. Cells were then incubated with primary antibodies in blocking solution (1% BSA/PBS) at 4 °C overnight and washed with 0.2% Tween20/PBS three times at 4 °C.

Antibodies were used against TRF2 (Novus, cat#NB110-57130, dilution 1:200) and XPF (Thermo Fisher Scientific, cat#MA5-12060, dilution 1:100), XPF (Bethyl Laboratories, cat# A301-315A, dilution 1:100) and ERCC1 (Santa Cruz Biotechnology, cat#sc-17809, dilution 1:100), RAP1 (Santa Cruz Biotechnology, cat#sc-28197, dilution 1:100) and BLM (Santa Cruz Biotechnology, cat#sc-365753, dilution 1:100), BRCA1 (Abcam, cat#ab16780, dilution 1:100), POLD3 (Abnova, cat# H00010714-M01, dilution 1:100), RAD51 (Abcam, cat#ab213, dilution 1:100), and RPA70 (Santa Cruz Biotechnology, cat#sc28304, dilution 1:100). After the incubation of secondary antibodies (in blocking solution) at RT for 1 h, and washed 3 times. Antibody information is provided in Supplementary Data 1. Images were captured using Olympus IX83 inverted microscopy with various Z-sections and then were compiled into 3D images to calculate the colocalization events.

## EdU incorporation assay

Cells were transfected with siRNA for 2 days, and RO-3306 (10 μM) was added and incubated for 21.5 h, followed by 10 μM EdU (5-Ethynyl-20-deoxyuridine) treatment for 1 h. For RCas9-sgRNA cell lines, cells were synchronized with thymidine and RO-3306[8]. Briefly, cells were first incubated with thymidine (2 mM) for 21 h, then released in fresh medium for 4 h, followed by RO-3306 (10 μM) treatment for 12 h. The EdU assay was detected using the Click-iT™ Plus EdU Cell Proliferation Kit (Thermo Fisher, Bremen, Germany; C10637), according to the manufacturer's instructions. Cells were fixed with 4% formaldehyde and permeabilized with 0.5% Triton X-100 in PBST for 10 min at 25 °C. After washing once with wash buffer, cells were then incubated with the Click-iT® Plus reaction cocktail at RT for 30 min. Cells were washed twice with 0.2% Tween 20 in TBS at RT for five minutes. After Click reactions, telomere DNA-FISH (TelG-Cy3) was performed as previously described. We only scored cells showing a punctuate big EdU staining (less than 30 EdU foci in the nucleus) and excluded S phase cells containing intensive pan-staining.

## Quantitative and statistical analyses for images

For immunostaining, 3D images were taken using an Olympus IX83 microscope and Hamamatsu C13440 digital camera. All images in the same experiments were captured with the same exposure time. The foci were selected by Imaris (Oxford Instruments) or cellSens (Olympus) software. Colocalization events were counted using the Imaris spot detection function in conjunction with the colocalization channel. For the quantification of telomere intensity, 3D images were projected to 2D with maximum intensity projection function, and the telomere foci were selected using the Imaris surface function. For quantification of TERRA intensity, TERRA signals were selected using cellSens. To calculate the intensity of XPF foci at telomeres, the signals of TRF2 foci were selected as ROI, the intensity of XPF was counted on the ROI of TRF2. Statistical significance determined by student's t-test or Mann-Whitney test and other parameters were analyzed by GraphPad Prism.

## siRNA knockdown

Cells were transfected with 5 nM of each siRNA using Lipofectamine RNAiMax (Thermo Fisher Scientific, cat#13778-150) in cells according to the manufacturer's instructions. siRNAs were purchased from Thermo Fisher Scientific (negative control siRNA 4390843, siXPF-1 s4800, siXPF-2 s4801, siXPG-1 s4802, siXPG-2 s4803, siFANCM s33621, siRNaseH1 s48357, siCSB s4805, siSLX4 s39053) or Dharmacon (siLuciferase, siFANCM, siBLM). Sequence information of Dharmacon siRNAs is provided in Supplementary Data 1. siRNA For FANCM and XPF double knockdown experiments, the final siRNA concentration was 10 nM (5 nM siFANCM plus 5 nM siXPF). In this case, 5 nM negative control siRNA plus 5 nM siRNA that targets FANCM or XPF were added into a single knockdown group respectively. For the long-term XPF knockdown experiments, cells were transfected with 10 nM of siNeg or siXPF (siXPF-1, s4800) for 3 days, and cells were cultured without

transfection reagents for another 4 days (week 1). Then the knockdown procedures were repeated for the next 7 weeks until week 8.

## Telomere restriction fragment analysis (TRF assay)

Genomic DNA was extracted by QIAamp DNA Mini Kit and step drew RNase treatment out 1 h to avoid RNA contamination. Genomic DNA (1 μg) was subjected to restriction digestion with RsaI and HinfI for 24 h at 37 °C, and resolved by agarose gel electrophoresis. After drying, gel was soaked in 0.5 M NaOH, 1.5 M NaCl for 20 min to denature DNA structure, then neutralized by soaking in 0.5 M Tris-HCl pH7.5, 1.5 M NaCl for 20 min. Dried gel was pre-hybridized by pre-warmed Church buffer (0.5 M NaPO4, 1 mM EDTA, 7% SDS, and 1% BSA in nuclease-free water) for 1 h at 50 °C and then hybridized overnight with $^{32}$P-labbel C-rich telomeric probe generated by RadPrimeTM DNA Labeling System (Thermo Fisher Scientific, cat#18428011) and purified with illustraTM MicroSpinTM G-25 Columns (GE Healthcare). After washing, gel was exposed in Phosphoimage film for 4 days and images were scanned by TyphoonTM FLA9000 biomolecular imager (GE Healthcare).

## RCas9 construction

The RCas9 vector was purchased from Addgene (LentiRCas9-CUG, #104183) and sgCUG sgRNA sequence was replaced by TERRA anti-sense or control sequences using NEBuilder HiFi DNA assembly (NEB, cat#E2621L). U2OS cells were transfected with RCas9 plasmids using Lipofectamine™ 2000 CD Transfection Reagent (Thermo Fisher Scientific, cat#12566014) and selected by 1 μg/ml puromycin for one week. Survival cell colonies were picked and maintained in 0.5 μg/ml puromycin DMEM medium. Positive clones were further confirmed by PCR. Cells were lysed in the lysis buffer (2X Roche PCR buffer, 0.45% NP 40, Tween 20) at 55 °C for 2 h and inactivated at 95 °C for 15 min. Cell lysates were subjected to PCR. Primers for RCas9 (forward primer (5'-GTT TAA GAG CTA TGC TGG AAA C-3'), reverse primer (5'-CCT AGC TAG CGA ATT CGC GC-3')) was used to detect 200 bp sequences located downstream of sgRNA. Plasmids and cell lines generated in this study are available from the corresponding author upon request.

## Viral transduction

HEK293T cells were co-transfected with RCas9-sgRNA plasmids, pCMVΔ8.91 and pDM VSV-G (ratio 1:0.1:0.9) using polyethylenimine (DNA: PEI = 12:50, 12 μg DNA for 4 × 10⁶ cells). Virus containing medium was collected after 48 h and 72 h post-transfection and further concentrated 10 times by PEG-8000. The concentrated virus pellet was resuspended in serum-free medium and stored at −80 °C. Cells (8 × 10⁴) were incubated with 1 mL of virus stock mixed with 1 mL of fresh medium and 2 μL of 10 μg/mL polybrene for 24 h. The virus containing medium was replaced with fresh medium and cells were cultured for another 24 h, and infected cells were selected by 1 μg/ml puromycin medium for 2 days. The cells were ready for EdU incorporation and the staining experiments.

## Drug sensitivity assay

Cells were seeded into 96-well microplates at a density of 3000–5000 cells/0.1 ml/well. After the cells were grown for 24 h, 0.1 ml of the culture medium containing various concentrations of doxorubicin (Pfizer), cisplatin (Hospira Australia Pty Ltd), etoposide (Fresenius Kabi Oncology Ltd), or NSC130813 (Sigma-Aldrich) was added to each well, and the treated cells were incubated for 72 h. Each concentration was tested in three wells. NSC130813 was pre-solubilized in DMSO prior to subsequent dilution in the medium. The final concentration of DMSO in the medium was less than 0.5%. After washing with PBS, the cells were fixed with cold methanol for 10 min on ice and stained with 0.5% crystal violet solution in 20% methanol for 10 min. After the plate was washed with water and dried, DMSO was added to the plate and the absorbance at 570 nm was measured by SpectraMax i3x (Molecular Devices). For XPF-ERCC1 staining, U2OS cells (1.5 × 10⁵ cells) were

seeded on the coverslips for 16 h and treated with 5 μM NSC130813 for 16 h.

## Ultraviolet light cross-linking-RNA immunoprecipitation (UV-RIP)

The UV-RIP protocol is described as the previous study[80]. Briefly, U2OS cells ($1.0 \times 10^8$ cells) were resuspended in 5 ml of ice-cold PBS, and irradiated with UV light at 200 mJ/cm$^2$ energy (254 nm). Cells were treated with 30 ml CSKT buffer (100 mM NaCl, 300 mM sucrose, 10 mM PIPES, 3 mM MgCl$_2$, 0.5% Triton X-100 in 1 L DEPC treated water, pH 6.8) for 15 min at 4 °C. After PBS wash once, UV-crosslinked cells ($1 \times 10^7$ cells) were treated with 200 μL of DNase I digestion buffer (50 mM Tris pH 7.5, 0.5% Nonidet P-40, 0.1% sodium lauroyl sarcosine, 600 U Turbo DNase I (ThermoFisher, Cat# AM2238), 80 U SuperaseIN (ThermoFisher, Cat#AM2696), 1x Protease inhibitors cocktail (Merck, Cat# 4693132001)) for 15 min at 37 °C. The proper amount of lysis buffer was added into the DNase I treated solution to reach the final concentration (1X PBS, 1% NP-40, 1% sodium deoxycholate, 160 U SuperaseIN, and 1x protease inhibitor cocktails) in 600 μL, and then was incubated at 4 °C for 25 min. Cell lysates were spun with 12,000 rpm for 10 min at 4 °C and the supernatant was collected. 10% lysate was collected as input. Antibodies (5 μg, IgG or XPF (Thermo-Fisher, Cat# MA5-12060)) were incubated with Dynabeads® Protein G (ThermoFisher) at 4 °C for 1 h. After two washes with DNase I buffer, antibody (Ab)-beads were added into cell lysates and incubated at 4 °C overnight. Beads were washed three times with ice-cold RIPA-I-200 wash buffer (50 mM HEPES, 10 mM EDTA (pH 8.0), 0.5% sodium deoxycholate, 0.5% NP-40 and 200 mM NaCl in DEPC-treated water). Following pre-washed once with ice-cold RIPA-II wash buffer (50 mM HEPES, 10 mM EDTA (pH 8.0), 0.5% Sodium Deoxycholate, 0.5% NP-40 and 50 mM NaCl in DEPC treated water), samples were treated with 30 U Turbo DNase I for 15 min at 37 °C. Ab-beads were washed twice with ice-cold RIPA-II wash buffer and eluted in 100 μl proteinase K buffer (100 mM Tris-HCl pH 7.5, 50 mM NaCl, 10 mM EDTA, 20 μg/μl Proteinase K, and 0.5% SDS) for 30 min at 55 °C.

## DNA-RNA immunoprecipitation (DRIP)

DRIP protocol was modified according to previous studies[83,84]. Cell pellets ($2 \times 10^6$ cells) were resuspended in nuclear extraction buffer (0.5% NP-40, 80 mM KCl, 0.5 mM HEPES pH 8.0) on ice for 30 min, and nuclei were spun down and resuspended in lysis buffer (1% SDS, 25 mM Tris-HCl, pH 8.0, 5 mM EDTA) for 30 min on ice and mild-sonicated for 2 min in ice-cold water. The cell lysates were incubated with proteinase K at 55 °C for 3 h. Genomic DNA was extracted by phenol-chloroform isoamyl alcohol (25:24:1) with the phase lock gel tubes and precipitated with ethanol. Genomic DNA was fragmented into 200-500 bp using Covaris S2 in microtubes with 10% duty cycle, 200 burst/cycles, intensity 3 for 60 s. For RNaseH controls, 8 μg of genomic DNA was treated with RNase H (5 U/μL, NEB, # M0523) at 37 °C overnight before immunoprecipitation, and purified by phenol-chloroform extraction. For immunoprecipitation, 5 μg genomic DNA in 250 μL of Tris-buffer (10 mM Tri-HCl, pH 8.0) was used per capture. 2 μg of S9.6 antibody (Millipore, # MABE1095) was conjugated to 20 μL Protein G Dynabeads (Thermo Fisher Scientific, #10004D) at 4 °C for 2 h. Genomic DNA was incubated with antibody-beads in 1X binding buffer prepared by adding 10X binding buffer (100 mM sodium phosphate, pH 7, 1.4 M NaCl, and 0.5% (vol/vol) Triton X-100) at 4 °C overnight. After twice washes with 1X binding buffer for 15 min at RT, immunoprecipitated DNA was eluted in 300 μL elution buffer (50 mM Tris, pH 8.0, 10 mM EDTA, pH 8.0, 0.5% (vol/vol) SDS) containing 7 μL of 20 mg/mL proteinase K at 55 °C for 45 min. DNA was purified and telomeric repeat DNA was measured by qPCR (forward primer: 5′-GGTTTTTGAGGGTGAGGGTG AGGGTGAGGGTGAGGGT-3′, reverse primer: 5′TCCCGACTATCCCTA TCCCTATCCCTATCCCTA-3′).

## Cell cycle analysis

Cells were fixed in 70% EtOH, stained with Propidium Iodide, and analyzed by flow cytometry. The gating strategy for flow cytometry is provided in Supplementary Data 2.

## Statistics and reproducibility

For immunostaining, foci number per cell was counted and the n depicted the number of cells or the number of telomere foci counted per group and the two-sided Mann-Whitney test (test for two populations of cells) was used for statistical analysis. For foci counting, all cells from several independent experiments were pooled for analysis shown in the figures. All data points of each independent experiment are provided in the Source Data file. Because the intensity scale of fluorescent images varies between different independent experiments, the absolute fluorescent intensities of foci were analyzed in each independent experiment independently, shown as a representative of three independent experiments; other independent experiments were provided in the Source Data file and showed the similar trends. For DRIP-qPCR, there are three technical replicates of qPCR for each biological replicate. The DRIP-qPCR result was one representative of three independent experiments, and all independent experiments showed similar trends provided in the Source data. For TRF assays, Student's $t$-test was calculated from three biological replicates. All raw data points and the plots of independent experiments were provided in the Source Data file.

## Reporting summary

Further information on research design is available in the Nature Research Reporting Summary linked to this article.

## Data availability

The data that support this study are available from the corresponding author upon reasonable request. TERRA iDRiP-MS proteomics data generated in this study have been deposited to the ProteomeXchange Consortium via the PRIDE with the identifier PXD028882. Source data are provided with this paper.

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

## Acknowledgements

We thank Dong Zhang, Jing-Jer, Lin and Shu-Chun Teng for valuable discussions. This work was supported by Ministry of Science and Technology (MOST) grants 109-2311-B-002-015-MY3 (H.-P.C.C.), 108-2628-B-002-004 (H.-P. C.C.), and National Taiwan University grants 109L891902 (H.-P.C.C.), NTU-AS-108L104305 (H.-P.C.C.), NTU-CC-107L892003 (H.-P.C.C.), NTU-CC-110L893402 (H.-P.C.C.), NTU-109L104045-1 (L.-W.C.) and 109L7880 (H.-P.C.C.). J.M.E. is a fellow of the Yushan Mountain Scholar project supported by the Minister of Education (MOE). Mass spectrometry data were acquired at the Academia Sinica Common Mass Spectrometry Facilities for Proteomics and Protein Modification Analysis located at the Institute of Biological Chemistry, Academia Sinica, supported by Academia Sinica Core Facility and Innovative Instrument Project (AS-CFII-108-107). We thank Technology Commons, College of Life Science, National Taiwan University for supporting the usage of equipment. We would like to thank the service provided by the Flow Cytometry Analyzing and Sorting Core of the First Core Laboratory, National Taiwan University College of Medicine.

## Author contributions

H.-P.C.C. conceived of and designed the study. C.-Y.G. conducted iDRiP, RNase H1 overexpression, and KD experiments. P.-C.C., L.-W.C., and T.-C.C. conducted KD experiments, western blotting and immuno-FISH staining. H.-J.S. conducted RCas9-sgRNA related experiments. I.-H.L. and C.-P.Y. performed DRIP qPCR. L.-Y.C. assisted TRF assay. C.-C.L. performed drug sensitivity assays. Y.C. performed UV-RIP. Y.-H.H. analyzed iDRiP data. Y.-Y.C. conducted quantitative mass spectrometry. T.W.-W.C. and J.-M.E. provides cell lines and advice on drug sensitivity experiments. C.-S.W. advised on DNA damage-related experiments. H.-P.C.C., C.-Y.G., H.-J.S., L.-W.C., and C.-C.L .wrote the manuscript.

## Competing interests

The authors declare no competing interests.
