## [Peer Review File · Nature Communications]

REVIEWER COMMENTS

Reviewer #1 (Remarks to the Author):

In the current manuscript, the authors describe a Rcas9-based system to degrade the long noncoding RNA TERRA in ALT cells. By isolating clonal cell lines expressing this system they show that TERRA supports the appearance of ALT features, including APBs, possibly telomeric R-loops and telomeric DNA synthesis outside of the S-phase. These data, although challenged by weaknesses associated with the experimental setup (see below) are in full agreement with a manuscript published months ago by the Azzalin laboratory (PMID: 34145295). In the second part of the paper the authors utilize iDRIP to identify TERRA interactors specifically in ALT cells and decide to focus on the endonuclease XPF. They show that XPF is enriched at ALT telomeres and that XPF localization at telomeres is negatively regulated by RNaseH1. Depletion of XPF reduces gammaH2AX accumulation at telomeres indicating that XPF is necessary for the formation of the physiological telomeric DNA damage typical of ALT cells and this is confirmed by double depletion of XPF and the translocase FANCM, which is known to induce massive DNA damage at ALT telomeres. Moreover, the authors show that depletion of XPF reduces accumulation of Brca1 and RAD51 at ALT telomeres as well as features of break induced replication (DNA synthesis in G2 and pOLD3 accumulation at telomeres). The authors also test the effects of inhibiting XPF activity on cell proliferation in ALT cells.

In my view, the most interesting part of the manuscript starts with the identification of XPF as a TERRA interactor in ALT cells. The first part of the manuscript is, in my opinion, based on weak experimental choices. The authors decide to isolate clonal cell lines expressing sgTERRA sequences and chose the two clones with lower TERRA levels for further characterization. Telomere length and TERRA expression are well known to vary among different clonal cell lines, and the different amounts of TERRA and telomere length among the different clones might derive from this clonal effect rather than TERRA depletion. Also, if the system was efficient, one would expect that all clones had lower TERRA compared to parental or control clones (which were not even analyzed here), but this seems not to be necessarily the case given the quite substantial variability in TERRA expression across sgTER clones. Overall, the system seems not to be established well enough and what the authors might be looking at could simply be clonal heterogeneity. Experiments should be performed with inducible clones or with populations. In addition, the above indicated manuscript from the Azzalin group has very well established that TERRA is a major trigger of ALT activity and demonstrated that TERRA drives APB formation and BIR in ALT cells. I thus fully disagree with the choice of the authors to pretend that these data are not out there. Credit has to be given to this work in the introduction and further discussed later on; it cannot solely be referenced to as some data consistent with the current work. If anything, it is the current work from the authors that is consistent with what has already been demonstrated by others.

The part on XPF should also be made more solid. The TERRA/XPF interaction should be validated by RIP in different cell lines and the analysis of the effects exerted by XPF depletion on ALT features should be

extended to several ALT cell lines. Also, the detection of telomeric R-loops is questionable as it is based on staining using the S9.6b antibody. The specificity of this antibody for microscopy has been challenged by many laboratories and no control (RNaseH treatment for example) is utilized here. DRIPs would be more solid, in my opinion. Also, I'm not sure I see, from the data presented here, how XPF drugging could represent a valid approach for ALT curing, the effects seem very mild and differences very variable across cell lines; this set of data is very weak. Finally, the XPF-mediated localization of RAD51 to ALT telomeres is interesting, however RAD51 has been shown to be dispensable for ALT-mediated telomere elongation by the Greenberg laboratory (PMID: 27760120); hence, the model proposed by the authors seems not to stand.

Reviewer #2 (Remarks to the Author):

In this manuscript Guh et al. claim that XPF endonuclease is recruited by TERRA R-loops at ALT telomeres and facilitates a DDR that will trigger break-induced replication through POLD3 loading. The paper has some interesting findings but there are major issues that need to be addressed to make this story suitable for publication.

Major comments:

1) Throughout their manuscript the authors use S9.6 immunofluorescence (IF) as the only method to assess R-loops. Over the past years there has been growing concerns over the validity of S9.6 IF due to its affinity to double stranded RNA (apart from RNA:DNA hybrids). Only recently, the lab of world leading expert on R-loops Frederic Chedin published a paper that addressed these concerns experimentally (DOI: 10.1083/jcb.202004079). The take home message was that without pretreatment of cells with RNaseT1 (that degrades RNA without affecting RNA:DNA hybrids) S9.6 IF is not reliable as a means to assess R-loops. Nevertheless, the use of S9.6 antibody in performing DRIP QPCR remains a valid method of R-loop assessment. Taking this into consideration and since the role of R-loops is heavily addressed throughout this study, I would ask the reviewers to include the proper pretreatments and controls in their S9.6 IF assays. In addition, I would suggest they reinforce their R-loop related findings with other methods with DRIP QPCR being the most reliable one.

2) The authors need to clarify how the WT and mutant RNaseH cells were constructed and assessed. Do these cells constantly or inducibly overexpress WT/mutant RNaseH? Is there any induction of a DNA damage response because of the overexpression of WT/mutant RNaseH?

Since the authors use these cells to address questions regarding telomeric R-loops it is very important that they validate the effect of these cell lines specifically on telomeric R-loops and not just on the global

R-loops. As mentioned before, the effect of these cells on R-loops needs to be addressed with other methods than S9.6 IF.

3) Regarding Figure 1h, the authors need to present an ethidium bromide gel in which they will have run the same amount of DNA as in the TRF analysis, as proof that the samples were equally loaded.

4) How did the authors validate the cell cycle synchronisation using the CDK1 inhibitor RO-3306? Also, in Figure 6d, 6e: how did the authors assess which cells were non-S?

5) In order to strengthen the statement that 'TERRA R-loops facilitate the recruitment of BLM and RPA to ALT telomeres' the authors need to assess how the WT and mutant RNaseH overexpression affect localization of BLM and RPA at telomeres

6) The authors make a point that their findings regarding XPF occur in ALT+ cells. It would be interesting to compare some of their key findings in ALT- cells. In particular, I would like to see how γ H2AX foci upon XPF depletion are affected in HeLa cells.

7) I understand that the role of FANCM in regulating telomeric R-loops has been shown by the Azzalin lab (DOI: 10.1038/s41467-019-10179-z) but the authors need to show this in their system. Therefore, in Figure 5b the induction of the telomeric R-loops in response to FANCM deletion needs to be challenged by WT RNaseH. Following that, the phenotypes of siFANCM-induced APBs (Figure 5c), siFANCM-induced γ H2AX (Figure 5e) and siFANCM-induced top5% telomere intensity (Figure 5f) also need to be challenged by WT RNaseH.

8) The authors need to indicate in the figure legends of each and every experiment the number of independent repeats that this was conducted.

Minor comments:

1) I don't understand what are the TERRA R-loop foci that are assessed in Fig 1g. Is the plot a different depiction of the analysis in plot Ext. Fig 1d? If this is the case it needs to be explained in the figure legend.

2) Regarding Extended Data Figure 1e-f, PLA is predominantly assessed by counting the number of PLA foci not the intensity of the PLA signal. I suggest that the authors include this measurement in their manuscript. Also, the label in the Y axis of the plot in Extended Data Fig 1f is misleading and needs to explain that this is PLA signal intensity and not telomeric R-loop foci intensity.

Reviewer #3 (Remarks to the Author):

In this manuscript, the authors study the role of XPF during Alternative Lengthening of Telomeres (ALT). First, the authors devise a CRISPR-based system to deplete TERRA RNA and find that in the absence of TERRA, both TERRA R-loops and ALT features are reduced. The authors then show that TERRA R-loops promote telomere clustering. They perform TERRA-iDRiP-MS to identify TERRA-interacting proteins and identify XPF. XPF co-localizes with TRF2 at telomeres in ALT cells in a TERRA R-loop dependent manner and is required for DNA break formation and RPA loading. Next, the authors show that depletion of FANCM leads to an increase in TERRA R-loops and DNA break formation. XPF (together with XPG) appears responsible for the breaks, though in a CSB and SLX4 independent manner. DSB formation by XPF promotes the recruitment of RAD51 and BRCA1 at telomeres, as well as of Pol delta for break-induced telomere synthesis.

Overall, this is an interesting paper reporting the novel finding that XPF promotes alternative lengthening of telomeres by acting on TERRA R loops. The experiments are sound and the results are convincing, although I do find that the effects are often very minimal and I therefore doubt what the real biological significance is. I think the authors should discuss this. Furthermore, I think that the authors could improve the manuscript by addressing the following concerns.

Major concerns

The authors propose that XPF promotes lengthening of telomeres in ALT cells, which is why these cells are more sensitive to XPF knockdown. To confirm this idea, they should show in a direct manner that telomeres are shortened upon loss of XPF in these cells, for instance using the TRF analysis also used after prolonged downregulation of TERRA shown in Fig 1h.

The authors show that XPF localization to telomeres depends on TERRA R-loop formation. They also show that the increase in gH2AX foci due to depletion of FANCM is dependent on XPF. Thus, they assume that XPF acts downstream of FANCM to generate DSB breaks, leading to recruitment of BRCA1 and RAD51. To strengthen this, they should show that localization of XPF to TERRA R-loops is increased after depletion of FANCM.

It is unexpected that if XPF processes telomeric R loops, its knockdown does not affect the amount of R loops (Fig 5). One would expect to observe more R loops, as is for instance observed after loss of XPG (Sollier et al, Mol Cell, 2014; Yasuhara et al, Cell, 2018). The authors should address this discrepancy.

Many reported effects appear only minimal. For instance, the reduction in number of APB and TERRA R loop foci (Fig 1f and g), the reduction in BLM colocalization (Fig 2g) the gH2AX reduction after siXPF (Fig 4f), the increase in TERRA R-loops after siFANCM (Fig 5b). This makes me doubt how widespread (at

telomeres) the identified mechanism, involving TERRA R loops and XPF, is. The effects appear significant but are they also biologically relevant? The authors should address this in the discussion.

In figure 1d, the authors show that depletion of TERRA RNA leads to reduced TRF2 foci (intensity). In figure 2g and h, they show that depletion of TERRA RNA reduces BLM and RPA70 colocalization with TRF2. Can this latter quantification be affected by the fact that TRF2 itself is already reduced by TERRA knockdown? Ideally, one should use another telomeric marker that is unaffected by TERRA knockdown.

lines 139-142. The authors compare cell lines (stably?) overexpressing mutant and wild type RNase H1. However, from the blot (extended figure 2g), it is not clear to what extent the ectopic RNase H1 is overexpressed compared to endogenous RNase H1. The authors should visualize endogenous RNase H1 as well. Also, the mutant RNase H1 is much more overexpressed than the wild type RNase H1. Therefore, to rule out that the observed differences between cells expressing either RNase H1 are due to the strong overexpression of mutant RNase H1 (which likely binds more strongly and prolonged to R loops and may therefore inhibit endogenous functional RNase H1), the authors should compare results shown in Figure 2k and extended Figure 2i also with U2OS cells not overexpressing any RNase H1.

At the end of the results section, the rationale behind testing doxorubicin, etoposide and cisplatin is unclear. It is interesting that ALT cells are more resistant to these drugs, but there can be many mechanisms (including DNA repair) responsible for this. If I understand correctly, I think the authors want to make the point that therefore it is interesting to have other treatment options for these cells, because of which they test the ERCC1-XPF inhibitor. However, this is not really clear to me and should be made clear. Also, it is not evident from extended Figure 7b that the ERCC1-XPF inhibitor affects ALT cells more than non-ALT cells, as only the HeLa cells appear an outlier. Thus, I do not agree with the final conclusion that based on these results XPF may be a valuable potential therapeutic target for ALT cancers. The authors should make clear why this is their conclusion based on these results. Also, they should show that indeed ERCC1/XPF is inhibited in HeLa and U2OS (and other) with the concentrations used, especially for instance the 4.5 μ M that shows a difference between HeLa and U2OS cells (in Figure 7b).

line 65. Xeroderma pigmentosum is not a progeroid syndrome of accelerated aging, but a cancer-prone and photosensitivity syndrome. ERCC1-XPF deficiency can indeed cause xeroderma pigmentosum but also other diseases such as xeroderma pigmentosum combined with the progeroid Cockayne syndrome and/or with the developmental disorder Fanconi anemia. Please rephrase and cite correctly to papers describing this.

Minor concerns

- Line 200. The authors mention that XPG knockdown or the double XPF/XPG knockdown reduces RPA foci but extended Figure 4 only shows gH2AX staining. Please also show the RPA staining and quantification.
- Extended fig 1E. How are the PLA assays validated for accuracy? Is this assay standard procedure in the lab and has it been determined, using proper control experiments (e.g. with single antibodies), that the foci truly represent R loop-TFR2 interactions?
- In line 237 the authors conclude that XPF promotes the recruitment of BRCA1 and RAD51 to ALT. I suggest to rephrase this such that it is clear that XPF (probably) not directly promotes BRCA1 and RAD51 recruitment (by interacting), but that it does so by generating a DNA break. The same suggestion for lines 259-260.
- XPF still co-localizes in a high percentage of TRF2 foci in non-ALT cells (Figure 4). What could this residual co-localization signify?
- Please avoid the use of expert abbreviations in the abstract like 'RCas9' and 'iDRiP-MS'
- I could not directly find how immunostainings such as shown in figure 4A were performed. Is this described in the methods?
- It is unclear to me why the PLA staining in extended Fig 1f was quantified by measuring the intensity of foci. Rather, I think that the authors should quantify the number of telomeric R-loop foci per cell.
- What does 'top 5% telomere foci intensity' means? Why is this quantified and not the complete telomeric intensity? Please explain this clearly in the text.
- In extended figure 2i it is unclear which groups are compared by the statistic test. It appears that all mutant RNase H1 cells are grouped and compared to all wild type RNase H cells grouped. However, the cells should be compared as separate groups.
- The authors write 'RNase H' but this should be 'RNase H1'

- Please add size markers to alle western blots

- Please clarify which type of cells are used for each of the different experiments (ALT proficient/deficient cells). Example: figure 6d.

- How do the results described in this study relate to reported observations that ERCC1/XPF induces telomere shortening?

REVIEWER COMMENTS

Reviewer #1 (Remarks to the Author):

In the current manuscript, the authors describe a Rcas9-based system to degrade the long noncoding RNA TERRA in ALT cells. By isolating clonal cell lines expressing this system they show that TERRA supports the appearance of ALT features, including APBs, possibly telomeric R-loops and telomeric DNA synthesis outside of the S-phase. These data, although challenged by weaknesses associated with the experimental setup (see below) are in full agreement with a manuscript published months ago by the Azzalin laboratory (PMID: 34145295). In the second part of the paper the authors utilize iDRiP to identify TERRA interactors specifically in ALT cells and decide to focus on the endonuclease XPF. They show that XPF is enriched at ALT telomeres and that XPF localization at telomeres is negatively regulated by RNaseH1. Depletion of XPF reduces gammaH2AX accumulation at telomeres indicating that XPF is necessary for the formation of the physiological telomeric DNA damage typical of ALT cells and this is confirmed by double depletion of XPF and the translocase FANCM, which is known to induce massive DNA damage at ALT telomeres. Moreover, the authors show that depletion of XPF reduces accumulation of Brca1 and RAD51 at ALT telomeres as well as features of break induced replication (DNA synthesis in G2 and polD3 accumulation at telomeres). The authors also test the effects of inhibiting XPF activity on cell proliferation in ALT cells.

In my view, the most interesting part of the manuscript starts with the identification of XPF as a TERRA interactor in ALT cells. The first part of the manuscript is, in my opinion, based on weak experimental choices. The authors decide to isolate clonal cell lines expressing sgTERRA sequences and chose the two clones with lower TERRA levels for further characterization. Telomere length and TERRA expression are well known to vary among different clonal cell lines, and the different amounts of TERRA and telomere length among the different clones might derive from this clonal effect rather than TERRA depletion. Also, if the system was efficient, one would expect that all clones had lower TERRA compared to parental or control clones (which were not even analyzed here), but this seems not to be necessarily the case given the quite substantial variability in TERRA expression across sgTER clones. Overall, the system seems not to be established well enough and what the authors might be looking at could simply be clonal heterogeneity. Experiments should be performed with inducible clones or with populations. In addition, the above indicated manuscript from the Azzalin group has very well established that TERRA is a major trigger of ALT activity and demonstrated that TERRA drives APB formation and BIR in ALT cells. I thus fully disagree with the choice of the authors to pretend that these data are not out there. Credit has to be given to this work in the introduction and further discussed later on; it cannot solely be referenced to as some data consistent with the current work. If anything, it is the current work from the authors that is consistent with what has already been demonstrated by others.

We thank the reviewer's comment pointing out that the interesting part is identifying XPF as a TERRA interacting protein. We introduced RCas9-sgTERRA using viral transduction and analyzed

gH2AX at telomeres, telomere intensity and EdU foci at telomeres in U2OS cells with mixed populations (New Fig. 6h, Supplementary Fig. 8a-8c) in the revised manuscript. The results are consistent with the observations from RCas9 clones, indicating that TERRA promotes break-induced telomere synthesis. These results also support that the RCas9-sgRNA system effectively knockdown TERRA transcripts (New Fig. 6h, Supplementary Fig. 8a-8c).

We apologize that we didn't explain Azzalin's findings clearly in the first submission, leading to some misunderstanding due to the wording. We added more about the findings from Azzalin's group in the introduction (Line 58-63) and discussion (Line 377-382). We did not want to pretend that their data are not out there and initially mentioned their work in the discussion. When we prepared this manuscript at the beginning, their paper (*Nat Comm*, 2021) was not out yet. We are sorry about the misunderstanding due to the wording, so we rephrased the sentences in the results (Line 301-302), and documented that our data are consistent with their results.

We selected clones instead of using cells with populations in the first place to analyze the long-term impact of TERRA depletion and to measure the telomere length by TRF assay. When culturing a mixed population of cells, highly proliferating cells will be selected after several passages. We speculated that TERRA depletion could have disadvantages for cell proliferation in ALT cells (TERRA depletion reduces telomere synthesis in ALT+ cells), and cells with low TERRA expression may grow slower and eventually be eliminated after long-term culturing. When introducing a transgene into the genome by random integration, the expression is controlled by not only its promoter but also its integration sites, which could be located at heterochromatin or euchromatin. There are also other epigenetic regulations that can further silence or activate the locus. Therefore, it is unlikely to get all clones with the same high expression of sgRNAs or RCas9, and low TERRA expression. Moreover, the Tet-induced system is recently called for caution because doxycycline can inhibit mitochondrial function (Johan Auwerx, 2015, Cell Report). The long-term treatment of doxycycline could have additional effects that are toxic to cells. Therefore, we think that selecting clones with lower TERRA expression is by far a better way to see the long-term impact of TERRA transcription on telomere length in ALT cells.

The results from Azzalin's group and our group both demonstrate that TERRA promotes ALT features and break-induced telomere synthesis. Here we further show that XPF is the critical factor to induce DNA breaks to activate telomere synthesis, and prolonged depletion of XPF reduced telomere length in U2OS cells (New Fig. 7a).

The part on XPF should also be made more solid. The TERRA/XPF interaction should be validated by RIP in different cell lines and the analysis of the effects exerted by XPF depletion on ALT features should be extended to several ALT cell lines.

We included the UV-RIP experiment in New Supplementary Fig. 3e and showed the interaction of XPF and TERRA in U2OS cells.

To analyze the role of XPF in other ALT cell lines, we depleted XPF in WI38-VA and found similar phenotypes as we observed in U2OS cells (New Fig. 6i). XPF depletion in FANCM-deficient cells attenuated EdU foci at telomeres in WI38-VA cells.

Also, the detection of telomeric R-loops is questionable as it is based on staining using the S9.6b antibody. The specificity of this antibody for microscopy has been challenged by many laboratories and no control (RNaseH treatment for example) is utilized here. DRIPs would be more solid, in my opinion.

We performed DRIP to confirm the telomeric R-loops in the revised version (New Fig. 5e, Supplementary Fig. 1g, 2h). The results from DRIP-qPCR are similar to the data from RNA-immunostaining (TERRA-TRF2-S9.6).

Also, I'm not sure I see, from the data presented here, how XPF drugging could represent a valid approach for ALT curing, the effects seem very mild and differences very variable across cell lines; this set of data is very weak.

Doxorubicin, etoposide and cisplatin all can induce DNA breaks to trigger cell death and usually are commonly used for the treatment of sarcomas. Sarcomas tend to utilize ALT for the extension of their telomeres (PMID: 29725455). And patients with ALT sarcomas showed a higher risk of death than patients with non-ALT sarcomas. We then wonder if the poor prognosis is because ALT cells are more resistant to conventional chemotherapy. Our results showed that ALT cells are more resistant to these drugs. The mechanism by which ALT cells are more resistant to conventional chemotherapy drugs is unclear and needed for further investigation. The NER pathway is reported to contribute to the drug resistance induced by platinum drugs (PMID:33291532, PMID:20418188, PMID:18090576) and XPF is one of the key components in the NER repair. Clinical studies have shown that higher expression of ERCC1 correlates with poor prognosis (PMID:1433335, PMID: 8040325), and lower expression of ERCC1 correlates with better responses to platinum drugs in patients (PMID:12114432). If ALT cells require XPF for telomere maintenance, the NER repair pathway should be very active in the normal condition. The higher NER activity in ALT cells could confer drug resistance. We added some sentences to discuss the idea (Line 454-459).

We think that XPF can be a therapeutic target because inhibition of XPF expression by siRNAs significantly reduces cell proliferation in ALT cells. And ALT cells tend to be resistant to conventional chemotherapy drugs. Indeed, our results only show that the XPF inhibitor NSC130813 is more toxic to HeLa than U2OS cells (Fig. 7c). It seems that ALT and non-ALT cell lines have similar sensitivity to NSC130813 (Supplementary Fig. 7b). It is also possible that NSC130813 could kill cancer cells in an XPF-independent manner. The mechanism of the toxicity caused by NSC130813 is not well-studied yet. ALT cancer cells are at least not more resistant to

NSC130813 compared to non-ALT cells. Therefore, inhibition of XPF could be an alternative for ALT cancer therapy. For treating cancer, we don't want to only specifically kill ALT cells but not non-ALT cells. The same toxicity to both ATL and non-ALT cells is good for clinical treatment. Targeting XPF is a proposed idea for ALT curing, but more studies are needed to find better or more specific drugs for it. We have rephrased the sentences in the result and the discussion (Line 328-338, Line 454-459) to restate our conclusion.

Finally, the XPF-mediated localization of RAD51 to ALT telomeres is interesting, however RAD51 has been shown to be dispensable for ALT-mediated telomere elongation by the Greenberg laboratory (PMID: 27760120); hence, the model proposed by the authors seems not to stand.

We have revised the model figure (New Fig. 7d). Although depletion of RAD51 didn't reduce the break-induced telomere synthesis, the recruitment of RAD51 is associated with increased telomere synthesis and homologous recombination (Fig. 6a-6c). There could be redundancy effects for break-induced synthesis. Other factors such as RAD52 or MRN complex can also contribute to that, thus single gene knockdown may not result in significant changes.

Reviewer #2 (Remarks to the Author):

In this manuscript Guh et al. claim that XPF endonuclease is recruited by TERRA R-loops at ALT telomeres and facilitates a DDR that will trigger break-induced replication through POLD3 loading. The paper has some interesting findings but there are major issues that need to be addressed to make this story suitable for publication.

We are thankful for the reviewer who thinks "our manuscript has some interesting findings" and thoughtful suggestions. We addressed the reviewer's questions point-by-point below.

Major comments:

1) Throughout their manuscript the authors use S9.6 immunofluorescence (IF) as the only method to assess R-loops. Over the past years there has been growing concerns over the validity of S9.6 IF due to its affinity to double stranded RNA (apart from RNA:DNA hybrids). Only recently, the lab of world leading expert on R-loops Frederic Chedin published a paper that addressed these concerns experimentally (DOI: 10.1083/jcb.202004079). The take home message was that without pretreatment of cells with RNaseT1 (that degrades RNA without affecting RNA:DNA hybrids) S9.6 IF is not reliable as a means to assess R-loops. Nevertheless, the use of S9.6 antibody in performing DRIP QPCR remains a valid method of R-loop assessment. Taking this into consideration and since the role of R-loops is heavily addressed throughout this study, I would ask the reviewers to include the proper pretreatments and controls in their S9.6 IF assays. In addition, I would suggest they reinforce their R-loop related findings with other methods with DRIP QPCR being the most reliable one.

We performed DRIP qPCR to analyze telomeric R-loops in the revised manuscript (New Fig. 5e, Supplementary Fig. 1g, 2h). The results are similar to the data obtained from TERRA R-loop staining (TERRA-S9.6-TRF2). The measurement of total R-loops using S9.6 antibody might be an issue in immunostaining as reported by Chedin's group. We usually see strong S9.6 signals at nucleoli where massive rRNA locate, but this is probably not a problem when measuring telomeric R-loops. We can observe clear S9.6 foci at telomeres, which are very different from the S9.6 staining in nucleoli.

2) The authors need to clarify how the WT and mutant RNaseH cells were constructed and assessed. Do these cells constantly or inducibly overexpress WT/mutant RNaseH? Is there any induction of a DNA damage response because of the overexpression of WT/mutant RNaseH?

Thanks for the suggestions. We mentioned that in the METHOD (Line 470-481). The cells (Supplementary Fig. 2g-2i) constantly express WT or mutant RNaseH-mCherry, or vector alone. U2OS cell lines with exogenous expression of WT RNase H1 and mutant RNase H1 (enzyme dead) were generated by transfection of pICE-RNaseH1-WT-NLS-mCherry (Addgene #60365) and pICE-RNaseH1-D10R-E48R-NLS-mCherry (Addgene #60367) and selected with puromycin.

We didn't observe big changes in gH2AX levels by western blotting between these cell lines (New Supplementary Fig. 2g), but a slightly increase in mutant RNaseH1 expressing cells. The data suggest that global DNA damage response might be slightly increased in cells expressing mutant RNaseH1-mCherry. We included the vector alone as a control, and gH2AX and endogenous RNase H1 levels are similar between cells expressing WT RNaseH1-mCherry and vector alone.

Since the authors use these cells to address questions regarding telomeric R-loops it is very important that they validate the effect of these cell lines specifically on telomeric R-loops and not just on the global R-loops. As mentioned before, the effect of these cells on R-loops needs to be addressed with other methods than S9.6 IF.

We performed DRIP qPCR to detect telomeric R-loops (New Fig. 5e, Supplementary Fig. 1g, 2h). TERRA knockdown (New Supplementary Fig. 1g) or overexpressing wildtype RNaseH1 (New Supplementary Fig. 2h) reduced telomeric R-loops detected by DRIP qPCR, while FANCM depletion increased telomeric R-loops (New Fig. 5e).

3) Regarding Figure 1h, the authors need to present an ethidium bromide gel in which they will have run the same amount of DNA as in the TRF analysis, as proof that the samples were equally loaded.

We provided the ethidium bromide gel in which we had run the same amount of DNA as in the TRF analysis in New Fig. 1h and Fig. 7a, as proof that the samples were equally loaded.

4) How did the authors validate the cell cycle synchronisation using the CDK1 inhibitor RO-3306? Also, in Figure 6d, 6e: how did the authors assess which cells were non-S?

We validated the cell cycle synchronization by flow cytometry (New Supplementary Fig. 7b, 7e), and cells in G2 phase increased upon synchronization (New Supplementary 7b,7e). In Figure 6d-6e, S phase and non-S phase cells are determined by the pattern of EdU signals when counting EdU foci at telomeres. S phase cells displayed a pan-staining of EdU signals with a great number of foci in the nucleus, and non-S phase cells contained no EdU or less than 30 big EdU foci in the nucleus as shown in New Supplementary Fig. 7c. We usually got around 30% of cells in S phase and 70% of cells in non-S phase after RO3306 treatment in Fig. 6d. This ratio was similar to the ratio determined by the flow cytometry. We described how we quantified the EdU foci in the method (Line 660). For RCas9-sgRNA cells, we synchronized cells with thymidine and RO3306 (New Supplementary Fig. 7e), and the flow cytometry showed that most cells were in G2 phase.

5) In order to strengthen the statement that 'TERRA R-loops facilitate the recruitment of BLM and RPA to ALT telomeres' the authors need to assess how the WT and mutant RNaseH overexpression affect localization of BLM and RPA at telomeres

To strengthen the statement that TERRA R-loops facilitate the recruitment of BLM and RPA to ALT telomeres, we depleted RNaseH1 in U2OS (New Fig. 2j-2n). We observed that the numbers of APBs, BLM, and RPA at telomeres were increased in RNase H1 knockdown cells, supporting that the R-loops affect the localization of BLM and RPA at telomeres.

6) The authors make a point that their findings regarding XPF occur in ALT+ cells. It would be interesting to compare some of their key findings in ALT- cells. In particular, I would like to see how γ H2AX foci upon XPF depletion are affected in HeLa cells.

We performed the staining for γ H2AX and telomere upon XPF depletion in HeLa cells (Reviewer's Figure 1). However, the colocalization of γ H2AX foci with telomeres was almost undetectable under the normal culture condition. The telomere foci are small and dispersed across the nucleus in HeLa cells. Because the ALT features such as telomere clustering and break-induced telomere synthesis are barely seen in non-ALT cells, it is very challenging to analyze these effects after XPF depletion in ALT- cells without any stress.

Reviewer's Figure 1. HeLa cells were transfected with siRNAs to knockdown XPF. The immunofluorescence (IF) was performed to detect γ H2AX (green) and telomeres (red).

7) I understand that the role of FANCM in regulating telomeric R-loops has been shown by the Azzalin lab (DOI: 10.1038/s41467-019-10179-z) but the authors need to show this in their system. Therefore, in Figure 5b the induction of the telomeric R-loops in response to FANCM deletion needs to be challenged by WT RNaseH. Following that, the phenotypes of siFANCM-induced APBs (Figure 5c), siFANCM-induced γ H2AX (Figure 5e) and siFANCM-induced top5% telomere intensity (Figure 5f) also need to be challenged by WT RNaseH.

We have provided evidence showing induction of WT RNase H1 reduced the phenotypes caused by FANCM deficiency (New Supplementary Fig. 6a-6c). We generated RNase H1 inducible system (Tet-On system), and the induction of WT RNase H1 expression reduced telomeric R-loops, APBs, γ H2AX at telomeres, and telomere intensity in FANCM-deficient cells.

8) The authors need to indicate in the figure legends of each and every experiment the number of independent repeats that this was conducted.

We have indicated the number of independent experiments in each experiment in the figure legends.

Minor comments:

1) I don't understand what are the TERRA R-loop foci that are assessed in Fig 1g. Is the plot a different depiction of the analysis in plot Ext. Fig 1d? If this is the case it needs to be explained in the figure legend.

Thanks for the thoughtful suggestion.

Yes, Fig. 1g is the quantification of TERRA R-loop foci in Ext. Fig 1d (now in Supplementary Fig. 1d). We have added the description in the figure legend in the revised manuscript.

2) Regarding Extended Data Figure 1e-f, PLA is predominantly assessed by counting the number of PLA foci not the intensity of the PLA signal. I suggest that the authors include this measurement in their manuscript. Also, the label in the Y axis of the plot in Extended Data Fig 1f is misleading and needs to explain that this is PLA signal intensity and not telomeric R-loop foci intensity.

We have relabeled the Y-axis and renamed it as S9.6-TRF2 PLA intensity (Now are Supplementary Fig. 1e-1f). We have included the quantification of the number and the intensity of the PLA foci per nucleus and added the description for quantification of PLA in the method (Line 670-671).

Reviewer #3 (Remarks to the Author):

In this manuscript, the authors study the role of XPF during Alternative Lengthening of

Telomeres (ALT). First, the authors devise a CRISPR-based system to deplete TERRA RNA and find that in the absence of TERRA, both TERRA R-loops and ALT features are reduced. The authors then show that TERRA R-loops promote telomere clustering. They perform TERRA-iDRiP-MS to identify TERRA-interacting proteins and identify XPF. XPF co-localizes with TRF2 at telomeres in ALT cells in a TERRA R-loop dependent manner and is required for DNA break formation and RPA loading. Next, the authors show that depletion of FANCM leads to an increase in TERRA R-loops and DNA break formation. XPF (together with XPG) appears responsible for the breaks, though in an CSB and SLX4 independent manner. DSB formation by XPF promotes the recruitment of RAD51 and BRCA1 at telomeres, as well as of Pol delta for break-induced telomere synthesis.

Overall, this is an interesting paper reporting the novel finding that XPF promotes alternative lengthening of telomeres by acting on TERRA R loops. The experiments are sound and the results are convincing, although I do find that the effects are often very minimal and I therefore doubt what the real biological significance is. I think the authors should discuss this. Furthermore, I think that the authors could improve the manuscript by addressing the following concerns.

We are grateful for the reviewer who found “this is an interesting paper reporting the novel finding that XPF promotes alternative lengthening of telomeres by acting on TERRA R loops”. We appreciate the reviewer’s thoughtful suggestions and address the reviewer’s concerns below.

The effects we observed from XPF, FANCM, and TERRA depletion are not minimal. The effects are often several-fold changes compared to control cells. We think that Alternative Lengthening of Telomeres involves dynamic events. The TERRA R-loops and DNA damage responses at ALT telomeres occur transiently and are resolved when break-induced telomere synthesis is completed. Thus, the number of TERRA R-loops or DNA breaks at telomeres seems to be low at a given time point in the normal condition. We have discussed this in the revised manuscript (Line 397-400).

Major concerns

The authors propose that XPF promotes lengthening of telomeres in ALT cells, which is why these cells are more sensitive to XPF knockdown. To confirm this idea, they should show in a direct manner that telomeres are shortened upon loss of XPF in these cells, for instance using the TRF analysis also used after prolonged downregulation of TERRA shown in Fig 1h.

Thank you for the suggestion.

We performed TRF analysis and showed that prolonged silencing XPF for 8 weeks in U2OS cells shortened the telomere length (New Fig. 7a) in the revised manuscript. The result supports that XPF is required for Alternative Lengthening of Telomeres.

The authors show that XPF localization to telomeres depends on TERRA R-loop formation. They also show that the increase in gH2AX foci due to depletion of FANCM is dependent on XPF.

Thus, they assume that XPF acts downstream of FANCM to generate DSB breaks, leading to recruitment of BRCA1 and RAD51. To strengthen this, they should show that localization of XPF to TERRA R-loops is increased after depletion of FANCM.

Thank you for the excellent suggestion. We have conducted the XPF, and TRF2 staining upon FANCM depletion and showed that FANCM depletion increased the colocalization events of XPF-TRF2 and the intensity of XPF foci that are colocalized with TRF2 (New Fig. 5a-5b), indicating that the recruitment of XPF to ALT telomeres is increased in FANCM knockdown cells. Although it is better to co-stain XPF, S9.6 and TRF2 to show the localization of XPF to TERRA R-loops, there is a technical difficulty due to the antibody issue. These antibodies are not compatible for three-color staining at the same time. As shown in Fig. 5c, the level of TERRA R-loops at telomeres is significantly elevated in FANCM knockdown cells. Altogether, these data imply that the localization of XPF to TERRA R-loops is increased upon FANCM depletion.

It is unexpected that if XPF processes telomeric R loops, its knockdown does not affect the amount of R loops (Fig 5). One would expect to observe more R loops, as is for instance observed after loss of XPG (Sollier et al, Mol Cell, 2014; Yasuhara et al, Cell, 2018). The authors should address this discrepancy.

We quantified the intensity and the number of S9.6 foci that are co-stained with TRF2 in the revised manuscript and found that XPF depletion increased the S9.6 intensity but not the number of foci in FANCM-deficient cells (New Fig. 5d). DRIP-qPCR also showed the level of telomeric R-loops was increased after loss of XPF in FANCM-deficient cells (New Fig. 5e). Loss of XPF doesn't affect the foci number of APBs, however, it affects the intensity of S9.6 that are colocalized with TRF2, suggesting that XPF is required for R-loop processing at ALT telomeres. Therefore, our result is consistent with previous findings. We rephrased our conclusion on Line 233-238.

Many reported effects appear only minimal. For instance, the reduction in number of APB and TERRA R loop foci (Fig 1f and g), the reduction in BLM colocalization (Fig 2g) the gH2AX reduction after siXPF (Fig 4f), the increase in TERRA R-loops after siFANCM (Fig 5b). This makes me doubt how widespread (at telomeres) the identified mechanism, involving TERRA R loops and XPF, is. The effects appear significant but are they also biologically relevant? The authors should address this in the discussion.

Those effects are actually robust but not minimal. There are several fold changes after FANCM, XPF or TERRA depletion. We now label the mean values of the foci number for the quantification in the graph to show the differences. TERRA R-loops and gH2AX events at telomeres are likely to appear transiently in ALT cells and the immunostaining only represents a snapshot at a given time point. Therefore, we only observed a fraction of cells displaying TERRA R-loops and gH2AX at telomeres by immunostaining. Telomere lengthening by telomerase is a dynamic event and the recruitment of telomerase to telomeres is driven by dynamic interactions (Thomas Cech, 2016, Cell). We believe that alternative lengthening of telomeres also involves dynamic interactions of different protein complexes and TERRA RNA with

telomeric DNA. The TERRA R-loops and DNA damage responses at ALT telomeres should occur transiently and be resolved when break-induced telomere synthesis is completed. We discussed the biological relevance of TERRA R-loops, FANCM, and XPF (Line 347-355, 383-400, 449-453) and explained these dynamic events in the discussion (Line 399-400).

The number of APBs reduces 2~5 fold in TERRA knockdown cells (Fig. 1f). The number of TERRA R-loop foci (TERRA-S9.6-TRF2) reduces 2~5 fold in TERRA knockdown cells (Fig. 1g). The number of BLM at big TRF2 foci reduces 5~6 fold after TERRA depletion (Fig. 1g).

The reduction of BLM at TRF2 (Fig. 2g) is 5~6 fold in TERRA knockdown cells.

The effect of gH2AX is reduced about 1.5~3 fold after XPF depletion (Fig. 4f, 5h) in single knockdown cells. In FANCM deficient cells, XPF knockdown reduces 2-fold of gH2AX foci at telomeres (Fig. 5h).

FANCM depletion causes a 5-fold increase (0.09 in control and 0.54 in siFANCM) in the number of TERRA R-loop foci (Fig. 5d).

In figure 1d, the authors show that depletion of TERRA RNA leads to reduced TRF2 foci (intensity). In figure 2g and h, they show that depletion of TERRA RNA reduces BLM and RPA70 colocalization with TRF2. Can this latter quantification be affected by the fact that TRF2 itself is already reduced by TERRA knockdown? Ideally, one should use another telomeric marker that is unaffected by TERRA knockdown.

We performed the RAP1 staining to mark telomeres in the revised manuscript (New Supplementary Fig. 2c). The results are similar to the data of co-staining with TRF2.

lines 139-142. The authors compare cell lines (stably?) overexpressing mutant and wild type RNase H1. However, from the blot (extended figure 2g), it is not clear to what extent the ectopic RNase H1 is overexpressed compared to endogenous RNase H1. The authors should visualize endogenous RNase H1 as well. Also, the mutant RNase H1 is much more overexpressed than the wild type RNase H1. Therefore, to rule out that the observed differences between cells expressing either RNase H1 are due to the strong overexpression of mutant RNase H1 (which likely binds more strongly and prolonged to R loops and may therefore inhibit endogenous functional RNase H1), the authors should compare results shown in Figure 2k and extended Figure 2i also with U2OS cells not overexpressing any RNase H1.

The cells were stably expressing RNase H1 (Now in Supplementary Fig. 2g-2i). We added the vector alone control (New Supplementary Fig. 2g-2i) not overexpressing RNase H1. The levels of endogenous RNase H1 are similar between these cell lines (New Supplementary Fig. 2g). In addition, we depleted RNase H1 by siRNAs and showed that the RNase H1 depletion results in a reduction of telomere clustering, APB formation, and the recruitment of BLM and RPA70 to telomeres (New Fig. 2j-2n) to support the idea of R-loops in the regulation of telomere clustering.

At the end of the results section, the rationale behind testing doxorubicin, etoposide and cisplatin is unclear. It is interesting that ALT cells are more resistant to these drugs, but there

can be many mechanisms (including DNA repair) responsible for this. If I understand correctly, I think the authors want to make the point that therefore it is interesting to have other treatment options for these cells, because of which they test the ERCC1-XPF inhibitor. However, this is not really clear to me and should be made clear. Also, it is not evident from extended Figure 7b that the ERCC1-XPF inhibitor affects ALT cells more than non-ALT cells, as only the HeLa cells appear an outlier. Thus, I do not agree with the final conclusion that based on these results XPF may be an valuable potential therapeutic target for ALT cancers. The authors should make clear why this is their conclusion based on these results. Also, they should show that indeed ERCC1/XPF is inhibited in HeLa and U2OS (and other) with the concentrations used, especially for instance the 4.5 μ M that shows a difference between HeLa and U2OS cells (in Figure 7b).

We analyzed the ERCC1/XPF interaction after the treatment of ERCC1-XPF inhibitor NSC130813, and showed that the colocalization events of ERCC1-XPF were decreased after the treatment of 5 μ M NSC130813 (New Supplementary Fig. 8f), supporting that ERCC1-XPF inhibitor NSC130813 can disrupt the interaction of ERCC1-XPF. However, the effect is only partial. Therefore, a better inhibitor of XPF-ERCC1 is needed for further investigation.

Doxorubicin, etoposide and cisplatin all can induce DNA breaks to trigger cell death and usually are commonly used for the treatment of sarcomas. Sarcomas tend to utilize ALT for the extension of their telomeres (PMID: 29725455). And patients with ALT sarcomas showed a higher risk of death than patients with non-ALT sarcomas. We then wonder if the poor prognosis is because ALT cells are more resistant to conventional chemotherapy. Our results showed that ALT cells are more resistant to these drugs. The mechanism by which ALT cells are more resistant to conventional chemotherapy drugs is unclear and needed for further investigation. The NER pathway is reported to contribute to the drug resistance induced by platinum drugs (PMID:33291532, PMID:20418188, PMID:18090576) and XPF is one of the key components in the NER repair. Clinical studies have shown that higher expression of ERCC1 correlates with poor prognosis (PMID:1433335, PIMD: 8040325), and lower expression of ERCC1 correlates with better responses to platinum drugs in patients (PMID:12114432). If ALT cells require XPF for telomere maintenance, the NER repair pathway should be very active in the normal condition. The higher NER activity in ALT cells could confer drug resistance. We added some sentences to discuss the idea (Line 454-459).

We think that XPF can be a therapeutic target because inhibition of XPF expression by siRNAs significantly reduces cell proliferation in ALT cells. And ALT cells tend to be resistant to conventional chemotherapy drugs. Indeed, our results only show that the XPF inhibitor NSC130813 is more toxic to HeLa than U2OS cells (Fig. 7c). It seems that ALT and non-ALT cell lines have similar sensitivity to NSC130813 (Supplementary Fig. 7b). It is also possible that NSC130813 could kill cancer cells in an XPF-independent manner. The mechanism of the toxicity caused by NSC130813 is not well-studied yet. ALT cancer cells are at least not more resistant to NSC130813 compared to non-ALT cells. Therefore, inhibition of XPF could be an alternative for ALT cancer therapy. For treating cancer, we don't want to only specifically kill ALT cells but not non-ALT cells. The same toxicity to both ATL and non-ALT cells is good for clinical treatment.

Targeting XPF is a proposed idea for ALT curing, but more studies are needed to find better or more specific drugs for it. We have rephrased the sentences in the result and the discussion (Line 328-338, Line 454-459) to restate our conclusion.

line 65. Xeroderma pigmentosum is not a progeroid syndrome of accelerated aging, but a cancer-prone and photosensitivity syndrome. ERCC1-XPF deficiency can indeed cause xeroderma pigmentosum but also other diseases such as xeroderma pigmentosum combined with the progeroid Cockayne syndrome and/or with the developmental disorder Fanconi anemia. Please rephrase and cite correctly to papers describing this.

Thanks for the suggestions. We rephrased the sentences and cited the references (Line 69-72).

Minor concerns

- Line 200. The authors mention that XPG knockdown or the double XPF/XPG knockdown reduces RPA foci but extended Figure 4 only shows gH2AX staining. Please also show the RPA staining and quantification.

The staining of RPA foci and the quantification are now in New Supplementary Fig. 5a-5b.

- Extended fig 1E. How are the PLA assays validated for accuracy? Is this assay standard procedure in the lab and has it been determined, using proper control experiments (e.g. with single antibodies), that the foci truly represent R loop-TFR2 interactions?

We included negative controls such as single antibody incubation in the New supplementary Figure 1e. Single antibodies (such as S9.6 alone, or TRF2 alone) didn't produce clear foci and showed no signals in each cell. Single antibodies always give a very low background and almost no foci that can be detected.

- In line 237 the authors conclude that XPF promotes the recruitment of BRCA1 and RAD51 to ALT. I suggest to rephrase this such that it is clear that XPF (probably) not directly promotes BRCA1 and RAD51 recruitment (by interacting), but that it does so by generating a DNA break. The same suggestion for lines 259-260.

Thanks for the suggestion. We rephrased those sentences in the revised manuscript (Line 265-268, 291).

- XPF still co-localizes in a high percentage of TRF2 foci in non-ALT cells (Figure 4). What could this residual co-localization signify?

There are several factors that can trigger the recruitment of XPF to telomeres. XPF-ERCC1 can recognize DNA junctions and cleaves DNA structures such as stem loops, bubbles, or flaps in one strand of a duplex (Neil McDonald, 2012, NAR). Here we show TERRA R-loops could be one of them. DNA secondary structures such as D-loop, and G-quadruplex structures in response to telomeric replication stress can also promote the recruitment of XPF to telomeres in non-ALT

cells in an R-loop independent manner. Thereby, these could result in XPF localization at telomeres in non-ALT cells.

- Please avoid the use of expert abbreviations in the abstract like 'RCas9' and 'iDRiP-MS'

We edited the abstract to avoid these abbreviations.

- I could not directly find how immunostainings such as shown in figure 4A were performed. Is this described in the methods?

We included the co-immunostaining method in the revised manuscript.

- It is unclear to me why the PLA staining in extended Fig 1f was quantified by measuring the intensity of foci. Rather, I think that the authors should quantify the number of telomeric R-loop foci per cell.

We now included the number and the intensity of PLA foci per nucleus in the New Supplementary Fig. 1e-1f. In ATL+ cells, telomeres tend to be clustered in APBs, the number of total telomere foci may be lower when big APBs are formed. Therefore, we quantified both the number and the intensity of PLA foci per cell. The results indicated that the number and intensity of PLA foci per nucleus were lower in TERRA knockdown cells.

- What does 'top 5% telomere foci intensity' means? Why is this quantified and not the complete telomeric intensity? Please explain this clearly in the text.

We quantified the complete telomere intensity for each spot (Fig. 2b-2c), and selected foci with the highest telomeric intensity (top 5 %) as telomere clustering events, in which several telomeres interact together at APBs or called ALT telomeres. The top 5% telomere intensity is usually around 6~10 fold higher (1976 a.u. in control and 1519 a.u. in TER1C6 cells) than the median of telomere intensity (244 a.u. in control and 151 a.u. in TER1C6 cells). We considered foci containing the highest telomere intensity as telomere clustering events. We also measured the size of telomere foci (Fig. 2e, Supplementary Fig 2a). The largest telomere foci (> 1.2 μm^2) account for 5% in control cells (sgSense and sgLamda2). These data indicate that TERRA depletion reduces the size (Fig. 2e, Supplementary Fig 2a) and the intensity (Fig. 2c-2d) of the telomere clustering hub.

We added more descriptions to explain the idea in the text (Line 126-135).

- In extended figure 2i it is unclear which groups are compared by the statistic test. It appears that all mutant RNase H1 cells are grouped and compared to all wild type RNase H cells grouped. However, the cells should be compared as separate groups.

We included a vector alone as a control, and overexpression of WT RNase H1 and mutant RNase H1 cells are compared to cells with vector alone (New Supplementary Fig. 2g-2i).

- The authors write 'RNase H' but this should be 'RNase H1'
Corrected. Thank you.

- Please add size markers to alle western blots
Done. Thank you.

- Please clarify which type of cells are used for each of the different experiments (ALT proficient/deficient cells). Example: figure 6d.
Thanks for the comment. We added the cell types in the figure legends.

- How do the results described in this study relate to reported observations that ERCC1/XPF induces telomere shortening?

It has been found by Xu-Dong Zhu's group that overexpressing wild type XPF induces telomere shortening and reduces the TRF2 association with telomeres in telomerase-positive (non-ALT) cells (PMID18812185). This effect is independent of XPF nuclease activity. It is unclear whether overexpressing XPF leads to increase the localization of XPF to telomeres or not in non-ALT cells.

It is an interesting question and we have not yet had good explanations. In ALT cells, the scenario is quite different. XPF is highly associated with telomeres. And the TRF2 also are enriched at telomeres. It is likely that the DNA repair and synthesis at telomeres are very different between ALT and non-ALT cells. The proteins that are associated with telomeres are also very different between ALT and non-ALT cells. The non-ALT telomeres rely on telomerase to extend the length, but ALT telomeres utilize the ALT mechanism that is independent of telomerase.

The biggest difference of telomeres between ALT and non-ALT cells is APB. PML associated bodies are typically formed in ALT cells, but not in non-ALT cells. Telomeres tend to be clustered in ALT cells, but not in non-ALT cells. Although the ERCC1/XPF could generate DNA breaks at telomeres in both ALT and non-ALT cells, the downstream of DNA damage response could be very dissimilar. It seems that telomeres in non-ALT cells prefer telomere fusion when telomeres are uncapped (Zhu, X.D et al., Mol Cell, 2003), but the ALT telomeres prefer homologous recombination and break-induced synthesis induced by the DNA damage response that is mediated by XPF. We added some sentences to point out the differences of XPF effect on telomere lengthening between ATL and non-ALT cells (Line 443-448).

REVIEWER COMMENTS

Reviewer #1 (Remarks to the Author):

The authors have performed a large number of experiments and rephrased several parts of the manuscript to address my concerns. I believe that this manuscript is now in a much better shape and I support its publication. It will be of great importance for the ALT and telomere fields.

Reviewer #2 (Remarks to the Author):

In this revised version of their manuscript Guh et al. have performed a lot of experiments to their work and addressed several my previous concerns. I find the paper has improved to a great extent. Nevertheless, there are some major concerns that were either not met or were revealed through this revision that need to be addressed before it is ready for publication.

Major comments:

1) Authors' response: We performed DRIP qPCR to analyze telomeric R-loops in the revised manuscript (New Fig. 5e, Supplementary Fig. 1g, 2h). The results are similar to the data obtained from TERRA R-loop staining (TERRA-S9.6-TRF2). The measurement of total R-loops using S9.6 antibody might be an issue in immunostaining as reported by Chedin's group. We usually see strong S9.6 signals at nucleoli where massive rRNA locate, but this is probably not a problem when measuring telomeric R-loops. We can observe clear S9.6 foci at telomeres, which are very different from the S9.6 staining in nucleoli.

Reviewer's comments: I thank the authors for performing the DRIP QPCR experiments. Nevertheless, they did not perform the RNaseT1 treatment on the coverslips of the S9.6 immunofluorescence (IF). I insist that without this control experiment any S9.6 IF data is unreliable. The excuse regarding the S9.6 IF that 'this is probably not a problem when measuring telomeric R-loops' is unscientific and needs to be experimentally challenged in order to be published in any scientific journal. Therefore, the authors should either perform experiments that they treat the coverslips with RNaseT1 before performing the S9.6 IF, or remove any S9.6 IF data completely from their manuscript.

2) Authors' response: Thanks for the suggestions. We mentioned that in the METHOD (Line 470-481). The cells (Supplementary Fig. 2g-2i) constantly express WT or mutant RNaseH-mCherry, or vector alone.

U2OS cell lines with exogenous expression of WT RNase H1 and mutant RNase H1 (enzyme dead) were generated by transfection of pICE-RNaseH1-WT-NLS-mCherry (Addgene #60365) and pICE-RNaseH1-D10R-E48R-NLS-mCherry (Addgene #60367) and selected with puromycin. We didn't observe big changes in gH2AX levels by western blotting between these cell lines (New Supplementary Fig. 2g), but a slightly increase in mutant RNaseH1 expressing cells. The data suggest that global DNA damage response might be slightly increased in cells expressing mutant RNaseH1-mCherry. We included the vector alone as a control, and gH2AX and endogenous RNase H1 levels are similar between cells expressing WT RNaseH1-mCherry and vector alone.

We performed DRIP qPCR to detect telomeric R-loops (New Fig. 5e, Supplementary Fig. 1g, 2h). TERRA knockdown (New Supplementary Fig. 1g) or overexpressing wildtype RNaseH1 (New Supplementary Fig. 2h) reduced telomeric R-loops detected by DRIP qPCR, while FANCM depletion increased telomeric R-loops (New Fig. 5e).

Reviewer's comments: I thank the authors for clarifying this and for performing the extra experiments.

3) Authors' response: We provided the ethidium bromide gel in which we had run the same amount of DNA as in the TRF analysis in New Fig. 1h and Fig. 7a, as proof that the samples were equally loaded.

Reviewer's comments: I thank the authors for performing this control experiment.

4) Authors' response: We validated the cell cycle synchronization by flow cytometry (New Supplementary Fig. 7b, 7e), and cells in G2 phase increased upon synchronization (New Supplementary 7b,7e). In Figure 6d-6e, S phase and non-S phase cells are determined by the pattern of EdU signals when counting EdU foci at telomeres. S phase cells displayed a pan-staining of EdU signals with a great number of foci in the nucleus, and non-S phase cells contained no EdU or less than 30 big EdU foci in the nucleus as shown in New Supplementary Fig. 7c. We usually got around 30% of cells in S phase and 70% of cells in non-S phase after RO3306 treatment in Fig. 6d. This ratio was similar to the ratio determined by the flow cytometry. We described how we quantified the EdU foci in the method (Line 660). For RCas9-sgRNA cells, we synchronized cells with thymidine and RO3306 (New Supplementary Fig. 7e), and the flow cytometry showed that most cells were in G2 phase.

Reviewer's comments: I thank the authors for performing this control experiments.

5) Authors' response: To strengthen the statement that TERRA R-loops facilitate the recruitment of BLM and RPA to ALT telomeres, we depleted RNaseH1 in U2OS (New Fig. 2j-2n). We observed that the numbers of APBs, BLM, and RPA at telomeres were increased in RNase H1 knockdown cells, supporting that the R-loops affect the localization of BLM and RPA at telomeres.

Reviewer's comments: I thank the authors for addressing my concern.

6) Authors' response: We performed the staining for γ H2AX and telomere upon XPF depletion in HeLa cells (Reviewer's Figure 1). However, the colocalization of γ H2AX foci with telomeres was almost undetectable under the normal culture condition. The telomere foci are small and dispersed across the nucleus in HeLa cells. Because the ALT features such as telomere clustering and break-induced telomere synthesis are barely seen in non-ALT cells, it is very challenging to analyze these effects after XPF depletion in ALT- cells without any stress.

Reviewer's comments: I thank the authors for performing this experiment.

7) Authors' response: We have provided evidence showing induction of WT RNase H1 reduced the phenotypes caused by FANCM deficiency (New Supplementary Fig. 6a-6c). We generated RNase H1 inducible system (Tet-On system), and the induction of WT RNase H1 expression reduced telomeric R-loops, APBs, γ H2AX at telomeres, and telomere intensity in FANCM-deficient cells.

Reviewer's comments: I thank the authors for performing these experiments. However, in Supplementary Figure 6c, the plot for telomere intensity shows no difference at all upon siFANCM but has a Mann-Whitbet test p value <0.0001 , also although the figure legend informs the reader that the plot contains means of the experiment, underneath the plot the median values are given. The authors need to check this plot for possible mistakes and correct it accordingly and if there is no difference, rewrite the text. Also, I refer the authors to the next comment (#8) regarding number of independent biological replicates of each experiment.

8) Authors' response: We have indicated the number of independent experiments in each experiment in the figure legends.

Reviewer's comments: I thank the authors for indicating the number of independent repeats of each experiment. However, there are still experiments that the number of independent repeats is not indicated, such as Fig. 1c-g, Fig. 2b, c, e, l-m, Fig. 5b, Fig 6.g, h, Suppl. Fig1. b-d, Suppl. Fig2a, c-g, l, etc.

Also, it is revealed that in most experiments performed in this study, the authors provide statistical analysis of experiments that involve only two independent biological repeats. This is plainly wrong, can lead to misleading conclusions and raises a lot of questions of the validity of those experiments. You need a minimum of three biological repeats in order to perform Student's t-test or Mann-Whitney test. The authors need to provide at least three biological repeats in all of their experiments.

Finally, I am perplexed with the phrase that the authors use in many of their legends ‘...independent experiments showed similar results.’ The authors just need to indicate the number of independent experiments that they performed and on which they applied their statistical analysis. This phrase implies that only the biological repeats that showed similar results are depicted and needs to be corrected everywhere in the text.

Minor comments:

1) Authors’ response: Thanks for the thoughtful suggestion. Yes, Fig. 1g is the quantification of TERRA R-loop foci in Ext. Fig 1d (now in Supplementary Fig. 1d). We have added the description in the figure legend in the revised manuscript.

Reviewer’s comments: I thank the authors for clarifying this.

2) Authors’ response: We have relabeled the Y-axis and renamed it as S9.6-TRF2 PLA intensity (Now are Supplementary Fig. 1e-1f). We have included the quantification of the number and the intensity of the PLA foci per nucleus and added the description for quantification of PLA in the method (Line 670-671).

Reviewer’s comments: I thank the authors for clarifying this.

Reviewer #3 (Remarks to the Author):

The authors have addressed most of my concerns in a satisfying manner. I still have few minor issues that I think they should address before publication. There are minor typo's, weird formulations and English grammar mistakes that should be corrected (for instance line 65: 'consisting of endonuclease activity'; line 131 'approximal'; line 137 'was' should be 'were'; Lines 397-400 etc.....). Also, I think the authors should consider the below remarks:

I do not agree with the author's conclusion that ALT cancer cells were more resistant to cisplatin (line 326). Supplementary Fig 8e shows that WI38 cells are just as sensitive as the non-ALT cells. The authors therefore cannot conclude that ALT cells are more resistant to 'conventional therapeutic drugs' (line 335), as they only observe this for doxorubicin and etoposide. This should be correctly stated. Also, their results in supplementary figure 8e suggest that ALT cells are not more sensitive than non-ALT cells to the NSC130813 XPF inhibitors. It is therefore weird that the authors end their results section by stating that 'suppressing XPF could be a potential treatment for ALT cancers' (line 337). Their own results suggest

that this is not likely. The same criticism holds true for lines 454-460, where the authors also come up with the unsubstantiated idea that consistent activation or higher activity of the NER pathway in ALT cells could be a means to confer drug resistance.

I agree with the authors who write in their rebuttal that a better inhibitor for XPF than NSC130813 is needed. I do not think that their result in supplemental Fig 8f, showing the 'co-localization' of ERCC1 with XPF is a valid read-out for XPF inhibition. For this, they should use a functional assay (testing XPF function). XPF could be functionally inhibited without affecting its interaction with ERCC1. If, however, its interaction with ERCC1 is inhibited, then both proteins will become unstable (as they form a dimer and stabilize each other) and likely not readily detectable with immunostaining. (van Vuuren, EMBO, 1993; Biggerstaff, EMBO, 1993). This is evidently not the case.

line 71. I suggest to write 'and similar segmental progeroid syndromes with severe features of accelerated aging' as both Cockayne syndrome and XFE syndrome are also segmental progeroid syndromes.

Line 78. Explain 'RCas9'

Reviewer #2 (Remarks to the Author):

In this revised version of their manuscript Guh et al. have performed a lot of experiments to their work and addressed several my previous concerns. I find the paper has improved to a great extent. Nevertheless, there are some major concerns that were either not met or were revealed through this revision that need to be addressed before it is ready for publication.

We are grateful for the reviewer's suggestions and pointing out that the paper has improved to a great extent. We addressed the reviewer's concerns below.

Major comments:

1) Authors' response: We performed DRIP qPCR to analyze telomeric R-loops in the revised manuscript (New Fig. 5e, Supplementary Fig. 1g, 2h). The results are similar to the data obtained from TERRA R-loop staining (TERRA-S9.6-TRF2). The measurement of total R-loops using S9.6 antibody might be an issue in immunostaining as reported by Chedin's group. We usually see strong S9.6 signals at nucleoli where massive rRNA locate, but this is probably not a problem when measuring telomeric R-loops. We can observe clear S9.6 foci at telomeres, which are very different from the S9.6 staining in nucleoli.

Reviewer's comments: I thank the authors for performing the DRIP QPCR experiments. Nevertheless, they did not perform the RNaseT1 treatment on the coverslips of the S9.6 immunofluorescence (IF). I insist that without this control experiment any S9.6 IF data is unreliable. The excuse regarding the S9.6 IF that 'this is probably not a problem when measuring telomeric R-loops' is unscientific and needs to be experimentally challenged in order to be published in any scientific journal. Therefore, the authors should either perform experiments that they treat the coverslips with RNaseT1 before performing the S9.6 IF, or remove any S9.6 IF data completely from their manuscript.

Thanks for the suggestion. We have removed the S9.6 staining data in the manuscript.

7) Authors' response: We have provided evidence showing induction of WT RNase H1 reduced the phenotypes caused by FANCM deficiency (New Supplementary Fig. 6a-6c). We generated RNase H1 inducible system (Tet-On system), and the induction of WT RNase H1 expression reduced telomeric R-loops, APBs, γ H2AX at telomeres, and telomere intensity in FANCM-deficient cells.

Reviewer's comments: I thank the authors for performing these experiments. However, in

Supplementary Figure 6c, the plot for telomere intensity shows no difference at all upon siFANCM but has a Mann-Whitbet test p value <0.0001 , also although the figure legend informs the reader that the plot contains means of the experiment, underneath the plot the median values are given. The authors need to check this plot for possible mistakes and correct it accordingly and if there is no difference, rewrite the text. Also, I refer the authors to the next comment (#8) regarding number of independent biological replicates of each experiment.

Thank you for the suggestion.

In the reversed manuscript, we showed the top 5 % telomere intensity in "New Supplementary Figure 6c". The effect on the top 5% was robust and significant.

We presented the mean values for foci number (Supplementary Figure 6c, APB foci and gH2AX foci), while we indicated the median values of the top 5% telomere intensity. We added the description in the figure legend.

8) Authors' response: We have indicated the number of independent experiments in each experiment in the figure legends.

Reviewer's comments: I thank the authors for indicating the number of independent repeats of each experiment. However, there are still experiments that the number of independent repeats is not indicated, such as Fig. 1c-g, Fig. 2b, c, e, l-m, Fig. 5b, Fig 6.g, h, Suppl. Fig1. b-d, Suppl. Fig2a, c-g, l, etc.

Also, it is revealed that in most experiments performed in this study, the authors provide statistical analysis of experiments that involve only two independent biological repeats. This is plainly wrong, can lead to misleading conclusions and raises a lot of questions of the validity of those experiments. You need a minimum of three biological repeats in order to perform Student's t-test or Mann-Whitney test. The authors need to provide at least three biological repeats in all of their experiments.

Finally, I am perplexed with the phrase that the authors use in many of their legends '...independent experiments showed similar results.' The authors just need to indicate the number of independent experiments that they performed and on which they applied their statistical analysis. This phrase implies that only the biological repeats that showed similar results are depicted and needs to be corrected everywhere in the text.

Thank you for your suggestions. We corrected the sentences. We conducted three independent experiments to confirm the data reproducibility and indicated the number of independent experiments in the figure legends. All biological replicates show similar results. Only the polit experiment that tested the TERRA knockdown efficiency of sgTERRA-1 and sgTERRA-2 was done once (Supplementary Figure 1a-1c). We had further validated the TERRA knockdown efficiency using sgTERRA-1 in the later experiments (Fig 1b-1d, Supplementary 8a).

The independent experiments were done at different times, and many of independent experiments were done by different people and the results were all reproducible. Because the images of each independent experiment were taken at different times, and the intensities were slightly different between independent experiments. Therefore, we analyzed data for each independent experiment independently. We calculated foci/per cell, and the n depicted the number of cells per group in one biological replicate and the Mann-Whitney test (test for two populations of cells and each cell has a value) was analyzed for each independent experiment. We presented one of independent experiments for foci counting in the figures. The detailed statistical analyses were described in Method section (Statistics and reproducibility). For DRIP-qPCR, there are three technical replicates (for qPCR) for each biological replicate (e.g. different KD experiments). The DRIP-qPCR result was one representative of three independent biological replicates. Student's t-test was calculated from three technical replicates of qPCR. For TRF assays, Student's t test was calculated from three biological replicates. All raw data points of all biological replicates are provided in the Source Data file.

Minor comments:

Reviewer #3 (Remarks to the Author):

The authors have addressed most of my concerns in a satisfying manner. I still have few minor issues that I think they should address before publication. There are minor typo's, weird formulations and English grammar mistakes that should be corrected (for instance line 65:'consisting of endonuclease activity'; line 131 'approximal'; line 137 'was' should be 'were'; Lines 397-400 etc.....). Also, I think the authors should consider the below remarks:

We appreciate the reviewer's comments and suggestions. We corrected the grammar mistakes and changed the wording.

I do not agree with the author's conclusion that ALT cancer cells were more resistant to cisplatin (line 326). Supplementary Fig 8e shows that WI38 cells are just as sensitive as the non-ALT cells. The authors therefore cannot conclude that ALT cells are more resistant to 'conventional therapeutic drugs' (line 335), as they only observe this for doxorubicin and etoposide. This should be correctly stated. Also, their results in supplementary figure 8e suggest that ALT cells are not more sensitive than non-ALT cells to the NSC130813 XPF inhibitors. It is therefore weird that the authors end their results section by stating that 'suppressing XPF could be a potential treatment for ALT cancers' (line 337). Their own results suggest that this is not likely. The same criticism holds true for lines 454-460, where the authors also come up with the unsubstantiated idea that consistent activation or higher

activity of the NER pathway in ALT cells could be a means to confer drug resistance.

Thank you for the suggestion. We restated the conclusion for the drug sensitivity (Line 318-321) and removed the sentences regarding to the higher activity of the NER pathway in ALT cells and suppressing XPF as a potential treatment.

The inhibition of XPF by NSC130813 kills both ALT and non-ALT cells. The data suggest that NSC130813 may have other targets rather than XPF and cause cell toxicity in non-ALT cells. Further studies are needed for the examination of NSC130813 and the development of more specific XPF inhibitors.

Silencing XPF by siRNAs results in telomere shortening and reduced growth rate in ALT cancer cells, thereby silencing XPF could be a potential treatment for ALT cancers. Although, further studies are needed for the development of XPF inhibitors, RNA therapy is now more realistic for clinical use.

I agree with the authors who write in their rebuttal that a better inhibitor for XPF than NSC130813 is needed. I do not think that their result in supplemental Fig 8f, showing the 'co-localization' of ERCC1 with XPF is a valid read-out for XPF inhibition. For this, they should use a functional assay (testing XPF function). XPF could be functionally inhibited without affecting its interaction with ERCC1. If, however, its interaction with ERCC1 is inhibited, then both proteins will become unstable (as they form a dimer and stabilize each other) and likely not readily detectable with immunostaining. (van Vuuren, EMBO, 1993; Biggerstaff, EMBO, 1993). This is evidently not the case.

Thanks for the suggestion. We performed the western blotting to check the levels of XPF and ERCC1 after NSC130813 treatment (New Supplementary Fig. 8g). The protein levels of XPF and ERCC1 were decreased, suggesting that those proteins became unstable.

line 71. I suggest to write 'and similar segmental progeroid syndromes with severe features of accelerated aging' as both Cockayne syndrome and XFE syndrome are also segmental progeroid syndromes.

Thank you for the suggestion. We added that in the sentence.

Line 78. Explain 'RCas9'

Thank you for the suggestion. We explained "RCas9" there.

REVIEWERS' COMMENTS

Reviewer #2 (Remarks to the Author):

Major comments:

1) Authors' response: We performed DRIP qPCR to analyze telomeric R-loops in the revised manuscript (New Fig. 5e, Supplementary Fig. 1g, 2h). The results are similar to the data obtained from TERRA R-loop staining (TERRA-S9.6-TRF2). The measurement of total R-loops using S9.6 antibody might be an issue in immunostaining as reported by Chedin's group. We usually see strong S9.6 signals at nucleoli where massive rRNA locate, but this is probably not a problem when measuring telomeric R-loops. We can observe clear S9.6 foci at telomeres, which are very different from the S9.6 staining in nucleoli.

Reviewer's comments: I thank the authors for performing the DRIP QPCR experiments. Nevertheless, they did not perform the RNaseT1 treatment on the coverslips of the S9.6 immunofluorescence (IF). I insist that without this control experiment any S9.6 IF data is unreliable. The excuse regarding the S9.6 IF that 'this is probably not a problem when measuring telomeric R-loops' is unscientific and needs to be experimentally challenged in order to be published in any scientific journal. Therefore, the authors should either perform experiments that they treat the coverslips with RNaseT1 before performing the S9.6 IF, or remove any S9.6 IF data completely from their manuscript.

Thanks for the suggestion. We have removed the S9.6 staining data in the manuscript.

Reviewer's comments: I thank the authors for doing this.

7) Authors' response: We have provided evidence showing induction of WT RNase H1 reduced the phenotypes caused by FANCM deficiency (New Supplementary Fig. 6a-6c). We generated RNase H1 inducible system (Tet-On system), and the induction of WT RNase H1 expression reduced telomeric R-loops, APBs, γ H2AX at telomeres, and telomere intensity in

FANCM-deficient cells.

Reviewer's comments: I thank the authors for performing these experiments. However, in Supplementary Figure 6c, the plot for telomere intensity shows no difference at all upon siFANCM but has a Mann-Whitbet test p value <0.0001 , also although the figure legend informs the reader that the plot contains means of the experiment, underneath the plot the median values are given. The authors need to check this plot for possible mistakes and correct it accordingly and if there is no difference, rewrite the text. Also, I refer the authors to the next comment (#8) regarding number of independent biological replicates of each experiment.

Thank you for the suggestion.

In the reversed manuscript, we showed the top 5 % telomere intensity in "New Supplementary Figure 6c". The effect on the top 5% was robust and significant.

We presented the mean values for foci number (Supplementary Figure 6c, APB foci and gH2AX foci), while we indicated the median values of the top 5% telomere intensity. We added the description in the figure legend.

Reviewer's comments: I thank the authors for doing this.

8) Authors' response: We have indicated the number of independent experiments in each experiment in the figure legends.

Reviewer's comments: I thank the authors for indicating the number of independent repeats of each experiment. However, there are still experiments that the number of independent repeats is not indicated, such as Fig. 1c-g, Fig. 2b, c, e, l-m, Fig. 5b, Fig 6.g, h, Suppl. Fig1. bd, Suppl. Fig2a, c-g, l, etc.

Also, it is revealed that in most experiments performed in this study, the authors provide statistical analysis of experiments that involve only two independent biological repeats. This is plainly wrong, can lead to misleading conclusions and raises a lot of questions of the validity of those experiments. You need a minimum of three biological repeats in order to

perform Student's t-test or Mann-Whitney test. The authors need to provide at least three biological repeats in all of their experiments.

Finally, I am perplexed with the phrase that the authors use in many of their legends '...independent experiments showed similar results.' The authors just need to indicate the number of independent experiments that they performed and on which they applied their statistical analysis. This phrase implies that only the biological repeats that showed similar results are depicted and needs to be corrected everywhere in the text.

Thank you for your suggestions. We corrected the sentences. We conducted three independent experiments to confirm the data reproducibility and indicated the number of independent experiments in the figure legends. All biological replicates show similar results. Only the polit experiment that tested the TERRA knockdown efficiency of sgTERRA-1 and sgTERRA-2 was done once (Supplementary Figure 1a-1c). We had further validated the TERRA knockdown efficiency using sgTERRA-1 in the later experiments (Fig 1b-1d, Supplementary 8a).

The independent experiments were done at different times, and many of independent experiments were done by different people and the results were all reproducible. Because the images of each independent experiment were taken at different times, and the intensities were slightly different between independent experiments. Therefore, we analyzed data for each independent experiment independently. We calculated foci/per cell, and the n depicted the number of cells per group in one biological replicate and the Mann-Whitney test (test for two populations of cells and each cell has a value) was analyzed for each independent experiment. We presented one of independent experiments for foci counting in the figures. The detailed statistical analyses were described in Method section (Statistics and reproducibility). For DRIP-qPCR, there are three technical replicates (for qPCR) for each biological replicate (e.g. different KD experiments). The DRIP-qPCR result was one representative of three independent biological replicates. Student's t-test was calculated from three technical replicates of qPCR. For TRF assays, Student's t test was calculated from three biological replicates. All raw data points of all biological replicates are provided in the Source Data file.

Reviewer's comments: I do not find the way the authors choose to apply statistics on their foci counting/intensity quantification-related plots and also on the way they choose to present them in their manuscript, to be scientifically correct.

The authors claim that they have done three independent repeats of each experiment and that all biological repeats show the same result. This is not depicted anywhere in the manuscript because the authors: 1) show a representative plot of only one of these independent experiments and 2) the statistical analysis that they show is of that one independent experiment.

This is misleading.

The reason that we perform three or more independent biological repeats of any experiment is to assess reproducibility and significance of the end result of all repeats. By applying statistics within one selected biological repeat, this gives us information on just that one, and therefore the question on the validity of all biological repeats is never addressed.

To give a hypothetical example of the importance of this. One scientist carries out three independent biological repeats to assess number of RPA foci between two conditions. First repeat shows a strong difference, second repeat shows a mild difference and the third shows no difference. If only the first repeat is presented and statistical analysis is applied only to that, then this is misleading from the overall truth.

Regardless if there were minor differences while carrying out the independent experiments (i.e. biological repeats), the authors need to either: 1) pool all their foci measurements of all three or more independent experiments together in plots, 2) calculate the mean or median of each independent experiment and then plot these. or 3) apply both of the previous comments in the same plot.

And once this is done, then the statistical analysis should be applied to all three independent experiments, not just to one of them.

I would kindly urge the authors to consult this article on plotting cell biology-related data:
<https://doi.org/10.1083/jcb.202001064>

Reviewer's comments: I do not find the way the authors choose to apply statistics on their foci counting/intensity quantification-related plots and also on the way they choose to present them in their manuscript, to be scientifically correct.

The authors claim that they have done three independent repeats of each experiment and that all biological repeats show the same result. This is not depicted anywhere in the manuscript because the authors: 1) show a representative plot of only one of these independent experiments and 2) the statistical analysis that they show is of that one independent experiment.

This is misleading.

The reason that we perform three or more independent biological repeats of any experiment is to assess reproducibility and significance of the end result of all repeats. By applying statistics within one selected biological repeat, this gives us information on just that one, and therefore the question on the validity of all biological repeats is never addressed.

To give a hypothetical example of the importance of this. One scientist carries out three independent biological repeats to assess number of RPA foci between two conditions. First repeat shows a strong difference, second repeat shows a mild difference and the third shows no difference. If only the first repeat is presented and statistical analysis is applied only to that, then this misleading from the overall truth.

Regardless if there were minor differences while carrying out the independent experiments (i.e. biological repeats), the authors need to either: 1) pool all their foci measurements of all three or more independent experiments together in plots, 2) calculate the mean or median of each independent experiment and then plot these. or 3) apply both of the previous comments in the same plot.

And once this is done, then the statistical analysis should be applied to all three independent experiments, not just to one of them.

I would kindly urge the authors to consult this article on plotting cell biology-related data: <https://doi.org/10.1083/jcb.202001064>

We thank the reviewer for the very informative suggestion.

We agree that it is better to present all independent experiments in the manuscript. We have combined all data points of three independent experiments for foci counting and performed the statistical analyses shown in the figures. The pooled data show similar results as independent experiments. The raw data points and the plot for each independent experiment are also provided in the Source Data file.

Because the scale of fluorescent intensity varies between independent experiments, it is unsuitable for combining the absolute fluorescent intensity altogether. Therefore, the plots of absolute telomere intensity are shown as representatives of three independent experiments. Other independent experiments are provided in the Source Data and show similar trends. We indicate that in the figure legend.